# KERNEL-BASED OPTIMALLY WEIGHTED CONFORMAL TIME-SERIES PREDICTION

**Jonghyeok Lee, Chen Xu & Yao Xie** *
H. Milton Stewart School of Industrial and Systems Engineering
Georgia Institute of Technology
Atlanta, GA 30332, USA
{jlee4177,cx9711}@gatech.edu, yao.xie@isye.gatech.edu

## ABSTRACT

In this work, we present a novel conformal prediction method for time-series, which we call Kernel-based Optimally Weighted Conformal Prediction Intervals (`KOWCPI`). Specifically, `KOWCPI` adapts the classic Reweighted Nadaraya-Watson (RNW) estimator for quantile regression on dependent data and learns optimal data-adaptive weights. Theoretically, we tackle the challenge of establishing a conditional coverage guarantee for non-exchangeable data under strong mixing conditions on the non-conformity scores. We demonstrate the superior performance of `KOWCPI` on real time-series against state-of-the-art methods, where `KOWCPI` achieves narrower confidence intervals without losing coverage.

## 1 INTRODUCTION

Conformal prediction, originated in Vovk et al. (1999; 2005), offers a robust framework explicitly designed for reliable and distribution-free uncertainty quantification. Conformal prediction has become increasingly recognized and adopted within the domains of machine learning and statistics (Lei et al., 2013; Lei & Wasserman, 2014; Kim et al., 2020; Angelopoulos & Bates, 2023). Assuming nothing beyond the exchangeability of data, conformal prediction excels in generating valid prediction sets under any given significance level, irrespective of the underlying data distribution and model assumptions. This capability makes it particularly valuable for uncertainty quantification in settings characterized by diverse and complex models.

Going beyond the exchangeability assumption has been a research challenge, particularly as many real-world datasets (such as time-series data) are inherently non-exchangeable. Tibshirani et al. (2019) addresses situations where a feature distribution shifts between training and test data and restores valid coverage through weighted quantiles based on the likelihood ratio of the distributions. More recently, Barber et al. (2023) bounds the coverage gap using the total variation distance between training and test data points and minimizes this gap using pre-specified data-independent weights. However, it remains open to how to appropriately optimize the weights.

To advance conformal prediction for time series, we extend the prior sequential predictive approach (Xu & Xie, 2023a;b) by incorporating nonparametric kernel regression into the quantile regression method on non-conformity scores. A key challenge of adapting this method to time-series data lies in selecting optimal weights to accommodate the dependent structure of the data. To ensure valid coverage of prediction sets, it is crucial to select weights inside the quantile estimator so that it closely approximates the true quantile of non-conformity scores.

In this paper, we introduce `KOWCPI`, which utilizes the Reweighted Nadaraya-Watson estimator (Hall et al., 1999) to facilitate the selection of data-dependent optimal weights. This approach anticipates that adaptive weights will enhance the robustness of uncertainty quantification, particularly when the assumption of exchangeability is compromised. Our method also addresses the weight selection issue in the weighted quantile method presented by Barber et al. (2023), as `KOWCPI` allows for the calculation of weights in a data-driven manner without prior knowledge about the data.

In summary, our main contributions are:

---

*Correspondence: yao.xie@isye.gatech.edu

- We propose `KOWCPI`, a sequential time-series conformal prediction method that performs nonparametric kernel quantile regression on non-conformity scores. In particular, `KOWCPI` learns optimal data-driven weights used in the conditional quantiles.
- We prove the asymptotic conditional coverage guarantee of `KOWCPI` based on the classical theory of nonparametric regression. We further obtain the marginal coverage gap of `KOWCPI` using the general result for the weights on quantile for non-exchangeable data.
- We demonstrate the effectiveness of `KOWCPI` on real time-series data against state-of-the-art baselines. Specifically, `KOWCPI` can achieve the narrowest width of prediction intervals without losing marginal and approximate conditional (i.e., rolling) coverage empirically.

## 1.1 LITERATURE

**RNW quantile regression** In Hall et al. (1999), the Reweighted Nadaraya-Watson (RNW, often referred to as Weighted or Adjusted Nadaraya-Watson) estimator was suggested as a method to estimate the conditional distribution function from time-series data. This estimator extends the renowned Nadaraya-Watson estimator (Nadaraya, 1964; Watson, 1964) by introducing an additional adjustment to the weights, thus combining the favorable bias properties of the local linear estimator with the benefit of being a distribution function by itself like the original Nadaraya-Watson estimator (Hall et al., 1999; Yu & Jones, 1998). The theory of the regression quantile with the RNW estimator has been further developed by Cai (2002). Furthermore, Cai (2002) and Salha (2006) demonstrated that the RNW estimator is consistent under strongly mixing conditions, which are commonly observed in time-series data. In this work, we adaptively utilize the RNW estimator within the conformal prediction framework to construct sequential prediction intervals for time-series data, leveraging its data-driven weights for quantile estimation and the weighted conformal approach.

**Conformal prediction with weighted quantiles** Approaches using quantile regression instead of empirical quantiles in conformal prediction have been widespread (Romano et al., 2019; Kivaranovic et al., 2020; Gibbs et al., 2023). These methods utilize various quantile regression techniques to construct conformal prediction intervals, and the convergence to the oracle prediction width can be shown under the consistency of the quantile regression function (Sesia & Candès, 2020). Another recent work by Guan (2023) uses kernel weighting based on the distance between the test point and data to perform localized conformal prediction, which further discusses the selection of kernels and bandwidths. Recent work in this direction of utilizing the weighted quantiles, including Lee et al. (2023), continues to be vibrant. As we will discuss later, our approach leverages techniques in classical non-parametric statistics when constructing the weights.

**Time-series conformal prediction** There is a growing body of research on time-series conformal prediction (Xu & Xie, 2021b; Gibbs & Candès, 2021). Various applications include financial markets (Gibbs & Candès, 2021), anomaly detection (Xu & Xie, 2021a), and geological classification (Xu & Xie, 2022). In particular, Gibbs & Candès (2021; 2024) sequentially construct prediction intervals by updating the significance level $\alpha$ based on the mis-coverage rate. This approach has become a major methodology for handling online, non-exchangeable data, leading to several subsequent developments of adaptively updating the single-parameter threshold that determines the prediction sets at each time step (Feldman et al., 2022; Auer et al., 2023; Bhatnagar et al., 2023; Zaffran et al., 2022; Angelopoulos et al., 2023; Yang et al., 2024; Angelopoulos et al., 2024). On the other hand, Xu & Xie (2023b); Xu et al. (2024) take a slightly different approach by conducting sequential quantile regression using non-conformity scores. Our study aims to integrate non-parametric kernel estimation for sequential quantile regression, addressing the weight selection issues identified by Barber et al. (2023). Additionally, our research aligns with Guan (2023), particularly in utilizing a dissimilarity measure between the test point and the past data.

## 2 PROBLEM SETUP

We begin by assuming that the observations of the random sequence $(X_t, Y_t) \in \mathbb{R}^d \times \mathbb{R}$, $t = 1, 2, \ldots$ are obtained sequentially. Notably, $X_t$ may represent exogenous variables that aid in predicting $Y_t$, the historical values of $Y_t$ itself, or a combination of both. (In Appendix A, we expand our discussion to include cases where the response $Y_t$ is multivariate.) A key aspect of our setup is that the data are

non-exchangeable and exhibit dependencies, which are typical in time-series data where temporal or sequential dependencies influence predictive dynamics.

Suppose we are given a pre-specified point predictor $\hat{f} : \mathbb{R}^d \to \mathbb{R}$ trained on a separate dataset or on past data. This predictor $\hat{f}$ maps a feature variable in $\mathbb{R}^d$ to a scalar point prediction for $Y_t$. Given a user-specified significance level $\alpha \in (0, 1)$, we use the initial $T$ observations to construct prediction intervals $\widehat{C}_{t-1}^\alpha(X_t)$ for $Y_t$ in a sequential manner from $t = T + 1$ onwards.

Two key types of coverage targeted by prediction intervals are *marginal coverage* and *conditional coverage*. Marginal coverage is defined as

$$\mathbb{P}(Y_t \in \widehat{C}_{t-1}^\alpha(X_t)) \geq 1 - \alpha, \tag{1}$$

which ensures that the true value $Y_t$ falls within the interval $\widehat{C}_{t-1}^\alpha(X_t)$ at least $100(1 - \alpha)\%$ of the time, averaged over all instances. On the other hand, conditional coverage is defined as

$$\mathbb{P}(Y_t \in \widehat{C}_{t-1}^\alpha(X_t) \mid X_t) \geq 1 - \alpha, \tag{2}$$

which is a stronger guarantee ensuring that given each value of predictor $X_t$, the true value $Y_t$ falls within the interval $\widehat{C}_{t-1}^\alpha(X_t)$ at least $100(1 - \alpha)\%$ of the time.

## 3 METHOD

In this section, we introduce our proposed method, KOWCPI (Kernel-based Optimally Weighted Conformal Prediction Intervals), which embodies our approach to enhancing prediction accuracy and robustness in the face of time-series data. We delve into the methodology and algorithm of KOWCPI in-depth, highlighting how the Reweighted Nadaraya-Watson (RNW) estimator integrates with our predictive framework.

Consider prediction for a univariate time series, $Y_1, Y_2, \ldots$. We have predictors $X_t$ given to us at time $t$, $t = 1, 2, \ldots$, which can depend on the past observations $(Y_{t-1}, Y_{t-2}, \ldots)$, and possibly other exogenous time series $Z_t$. Given a pre-trained algorithm $\hat{f}$, we also have a sequence of non-conformity scores indicating the accuracy of the prediction:

$$\hat{\varepsilon}_t = Y_t - \hat{f}(X_t), \quad t = 1, 2, \ldots.$$

We denote the collection of the past $T$ non-conformity scores at time $t > T$ as

$$\mathcal{E}_t = (\hat{\varepsilon}_{t-1}, \ldots, \hat{\varepsilon}_{t-T}),$$

We construct the prediction interval $\widehat{C}_{t-1}^\alpha(X_t)$ with significance level $\alpha$ at time $t > T$ as follows:

$$\widehat{C}_{t-1}^\alpha(X_t) = [\hat{f}(X_t) + \hat{q}_{\beta^*}(\mathcal{E}_t), \hat{f}(X_t) + \hat{q}_{1-\alpha+\beta^*}(\mathcal{E}_t)],$$
$$\beta^* = \underset{\beta \in [0, \alpha]}{\operatorname{argmin}} \left( \hat{q}_{1-\alpha+\beta}(\mathcal{E}_t) - \hat{q}_\beta(\mathcal{E}_t) \right). \tag{3}$$

Here, $\hat{q}_\beta$ is a quantile regression algorithm that returns an estimate of the $\beta$-quantile of the residuals, which we will explain through this section. We consider asymmetrical confidence intervals to ensure the tightest possible coverage.

### 3.1 REWEIGHTED NADARAYA-WATSON ESTIMATOR

The Reweighted Nadaraya-Watson (RNW) estimator is a general and popular method for quantile regression for time series. Observe $(\tilde{X}_i, \tilde{Y}_i)$, $i = 1, \ldots, n$, where $\tilde{Y}_i \in \mathbb{R}$, and the predictors $\tilde{X}_i$ can be $p$-dimensional. The goal is to predict the quantile $\mathbb{P}(\tilde{Y} \leq b|\tilde{X})$, $b \in \mathbb{R}$, given a test point $\tilde{X}$ using training samples. The RNW estimator introduces *adjustment weights* on the predictors to ensure consistent estimation. We define the probability-like adjustment weights $p_i(\tilde{X})$, $i = 1, \ldots, n$, by maximizing the empirical log-likelihood $\sum_{i=1}^n \log p_i(\tilde{X})$, subject to $p_i(\tilde{X}) \geq 0$, and

$$\sum_{i=1}^n p_i(\tilde{X}) = 1, \tag{4}$$

$$\sum_{i=1}^n p_i(\tilde{X})(\tilde{X}_i - \tilde{X})K_h(\tilde{X}_i - \tilde{X}) = 0. \tag{5}$$

The RNW estimate of the conditional CDF $\mathbb{P}(\tilde{Y} \leq b | \tilde{X})$ is defined as follows:

$$\widehat{F}(b | \tilde{X}) = \sum_{i=1}^{n} \widehat{W}_i(\tilde{X}) \mathbb{1}(\tilde{Y}_i \leq b), \tag{6}$$

where the final weights $\widehat{W}_i$ are given by

$$\widehat{W}_i(\tilde{X}) = \frac{p_i(\tilde{X}) K_h(\tilde{X}_i - \tilde{X})}{\sum_{i=1}^{n} p_i(\tilde{X}) K_h(\tilde{X}_i - \tilde{X})}, \quad i = 1, \ldots, n, \tag{7}$$

computed as a weighted average of the adjustment weights $p_i$ based on the similarity between $\tilde{X}$ to each sample $\tilde{X}_i$ measured by the kernel function $K \colon \mathbb{R}^p \to \mathbb{R}$. Here, $K_h(u) = h^{-p} K(h^{-1} u)$ for $u \in \mathbb{R}^p$. Any reasonable choice of kernel function is possible; however, to ensure the validity of our theoretical results discussed in Section 4, the kernel should be nonnegative, bounded, continuous, and possess compact support. An example is $K(u) = k(\|u\|)$, where $k \colon \mathbb{R} \to \mathbb{R}$ is the Epanechnikov kernel.

The primary computational burden of the RNW estimator lies in calculating the adjustment weights $p_i$. However, as Lemma 3.1 shows, this reduces to a simple one-dimensional convex minimization problem, ensuring that the RNW estimator is not computationally intensive. This simplification significantly alleviates the overall computational complexity. Furthermore, Lemma 3.1 serves as a starting point for the proof of the asymptotic conditional coverage property of our algorithm, which will be addressed in Appendix B.1.

**Lemma 3.1** ((Hall et al., 1999; Cai, 2001)). *The adjustment weights $p_i(\tilde{X})$, $i = 1, \ldots, n$, for the RNW estimator are given as*

$$p_i(\tilde{X}) = \frac{1}{n} \left[ 1 + \lambda([\tilde{X}_i]_1 - [\tilde{X}]_1) K_h(\tilde{X}_i - \tilde{X}) \right]^{-1}, \tag{8}$$

*where $[X]_1$ denotes the first element of a vector $X$, and $\lambda \in \mathbb{R}$ is the minimizer of:*

$$L(\lambda; \tilde{X}) = -\sum_{i=1}^{n} \log\{1 + \lambda([\tilde{X}_i]_1 - [\tilde{X}]_1) K_h(\tilde{X}_i - \tilde{X})\}. \tag{9}$$

### 3.2 RNW FOR CONFORMAL PREDICTION

To perform the quantile regression for prediction interval construction at time $t = T + 1$, we use a sliding window approach, breaking the past $T$ residuals $\mathcal{E}_{T+1} = (\hat{\varepsilon}_T, \ldots \hat{\varepsilon}_1)$ into $n := (T - w)$ overlapping segments of length $w$. We construct the predictors and responses to fit the RNW estimator as follows:

$$\tilde{Y}_i = \hat{\varepsilon}_{i+w}, \quad \tilde{X}_i = (\hat{\varepsilon}_{i+w-1}, \ldots, \hat{\varepsilon}_i), \quad i = 1, \ldots, n.$$

With RNW estimator $\widehat{F}$ fitted on $((\tilde{X}_i, \tilde{Y}_i))_{i=1}^{n}$, the conditional $\beta$-quantile estimator $\widehat{Q}_\beta$ is defined as

$$\widehat{Q}_\beta(\tilde{X}) := \inf\{\tilde{y} \in \mathbb{R} \colon \widehat{F}(\tilde{y} | \tilde{X}) \geq \beta\}. \tag{10}$$

After time $t = T + 1$, we update $\mathcal{E}_t$ by removing the oldest residual and adding the newest one, then repeat the process (see Algorithm 1). In Section 4, we prove that due to the consistency of $\widehat{Q}_\beta$, KOWCPI achieves asymptotic conditional coverage despite the significant temporal dependence introduced by using overlapping segments of residuals.

*Bandwidth selection.* Estimating the theoretically optimal bandwidth that minimizes the asymptotic mean-squared error requires additional derivative estimation, which significantly complicates the problem. Consequently, similar to general non-parametric models, one can use cross-validation to select the bandwidth. However, cross-validation can be computationally burdensome and may deteriorate under dependent data (Fan et al., 1995). Therefore, we adapt the non-parametric AIC (Cai & Tiwari, 2000), used for bandwidth selection in local linear estimators. This method is applicable because the RNW estimator belongs to the class of linear smoother (Cai, 2002). Let $S$ be a linear smooth operator, with the $(i, j)$-th element given by $S_{ij} = \widehat{W}_j(\tilde{X}_i)$ (Hastie, 1990). Recognizing that

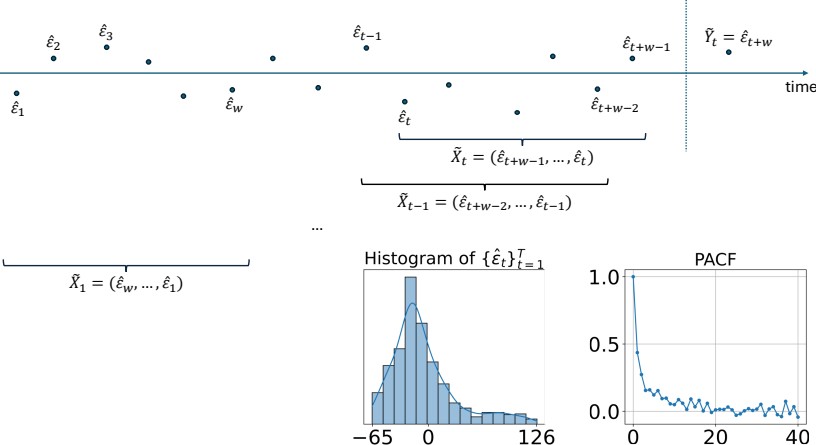

Figure 1: Illustration of KOWCPI, a sequential conformal prediction method. In the absence of exchangeability in the data, as indicated by the empirical distribution of residuals and the PACF plot, it is critical to consider the sequentially dependent structure of the data. In KOWCPI, non-conformity score blocks are updated sequentially using a sliding window, which provides prediction intervals for future scores through nonparametric quantile regression.

---

**Algorithm 1** Kernel-based Optimally Weighted Conformal Prediction Intervals (KOWCPI)

---

**Require:** Training data $(X_t, Y_t)$, $t = 1, \ldots, T$, pre-trained point predictor $\hat{f}$, target significance level $\alpha \in (0, 1)$, window length $w$, non-conformity score block count $n = T - w$.

1: Compute prediction residuals for the training data: $\hat{\varepsilon}_t = Y_t - \hat{f}(X_t)$, $t = 1, \ldots, T$.
2: **for** $t = T + 1, T + 2, \ldots$ **do**
3:     Update residual history $\mathcal{E}_t = (\hat{\varepsilon}_{t-1}, \ldots, \hat{\varepsilon}_{t-T})$.
4:     Break $\mathcal{E}_t$ into overlapping segments: $\tilde{X}_i = (\hat{\varepsilon}_{t-T+i+w-2}, \ldots, \hat{\varepsilon}_{t-T+i-1})$, $i = 1, \ldots, n + 1$.
5:     Compute $\lambda$ that minimizes $L(\cdot; \tilde{X}_{n+1})$ in equation 9.
6:     Derive adjustment weights $p_i(\tilde{X}_{n+1})$, $i = 1, \ldots, n$, and calculate final weights $\widehat{W}_i(\tilde{X}_{n+1})$ in equation 7.
7:     Using $\tilde{Y}_i = \hat{\varepsilon}_{t-T+i+w-1}$, $i = 1, \ldots, n$, compute the quantile estimator $\widehat{Q}_\beta(\tilde{X}_{n+1})$ for $\beta \in [0, \alpha]$.
8:     Determine $\beta^* = \operatorname{argmin}_{\beta \in [0,\alpha]} \widehat{Q}_{1-\alpha+\beta}(\tilde{X}_{n+1}) - \widehat{Q}_\beta(\tilde{X}_{n+1})$.
9:     Return prediction interval $\widehat{C}_{t-1}^\alpha(X_t) = \left[ \hat{f}(X_t) + \widehat{Q}_{\beta^*}(\tilde{X}_{n+1}), \ \hat{f}(X_t) + \widehat{Q}_{1-\alpha+\beta^*}(\tilde{X}_{n+1}) \right]$.
10:    Obtain new residual $\hat{\varepsilon}_t = Y_t - \hat{f}(X_t)$.
11: **end for**

---

the degree of freedom of the RNW smoother can be given as $\operatorname{tr}(SS^\top)$, we choose the bandwidth $h$ that minimizes

$$\mathrm{AIC}_C(h) := \log(\mathrm{RSS}) + \frac{n + \operatorname{tr}(SS^\top)}{n - (\operatorname{tr}(SS^\top) + 2)}, \tag{11}$$

where RSS is the residual sum of squares.

*Window length selection.* To select the window length $w$, cross-validation can be employed. In experiments, we chose $w$ with the smallest average width that achieves a target coverage in the validation set. Another approach is to use a weighted sum of the average under-coverage rate and the average width obtained for a given $w$ as the criterion. We note that the performance is generally less sensitive to the choice of $w$ across a broader range compared to the bandwidth $h$. Additionally, in Appendix E, we introduce an adaptive window selection approach that enables $w$ to be determined in a data-driven manner, eliminating the need for hyperparameter tuning with minimal loss in performance.

## 4 THEORY

In this section, we introduce the theoretical properties of the RNW estimator, a quantile regression method we use, and demonstrate in Theorem 4.9 that our KOWCPI asymptotically displays conditional coverage under the strong mixing of residuals. It turns out that the asymptotic conditional coverage gap can be derived from known results in the context of kernel quantile regression.

### 4.1 MARGINAL COVERAGE

We begin by bounding the marginal coverage gap of the KOWCPI method. The following result shows the coverage gap using our weights, compared with the oracle weights; the results are established using a similar strategy as in Tibshirani et al. (2019, Lemma 3):

**Proposition 4.1** (Non-asymptotic marginal coverage gap). *Denote by $\mathcal{P}$ the joint density of $(\tilde{Y}_1, \ldots, \tilde{Y}_{n+1})$. Then, we have*

$$\left| \mathbb{P}\left(Y_{T+1} \in \widehat{C}_T^\alpha(X_{T+1})\right) - (1-\alpha) \right| \leq \frac{1}{2}\left(\mathbb{E}\left\|(W^*)_{1:n} - \widehat{W}\right\|_1 + \mathbb{E}W_{n+1}^*\right) + 2\mathbb{E}\Delta(\tilde{X}_{n+1}) \quad (12)$$

*where $\Delta(\tilde{X}_{n+1})$ is the discrete gap defined in equation 17, and $W^*$ is the vector of oracle weights with each entry defined as*

$$W_i^* := \frac{\sum_{\sigma:\sigma(n+1)=i} \mathcal{P}(\tilde{Y}_{\sigma(1)}, \ldots, \tilde{Y}_{\sigma(n+1)})}{\sum_\sigma \mathcal{P}(\tilde{Y}_{\sigma(1)}, \ldots, \tilde{Y}_{\sigma(n+1)})}, \quad i = 1, \ldots, n+1, \quad \text{(oracle weights)} \quad (13)$$

*and $\sigma$ is a permutation on $\{1, \ldots, n+1\}$.*

The implication of Proposition 4.1 is that

- The "under-coverage" depends on the $\ell_1$-distance between the learned optimal weights and oracle-optimal weights (that depends on the true joint distribution of data).

- Note that the oracle weights $W_t^*$ cannot be evaluated, because in principle, it requires considering the $(n+1)!$ possible shuffled observed residuals and their joint distributions.

- The form of the oracle weights $W_i^*$ from equation 13 offers an intuitive basis for algorithm development: we can practically estimate the weights through quantile regression, utilizing previously observed non-conformity scores.

### 4.2 CONDITIONAL COVERAGE

In this section, we derive the asymptotic conditional coverage property of KOWCPI. For this, we introduce the assumptions necessary for the consistency of the RNW estimator. To account for the dependency in the data, we assume the strong mixing of the residual process.

A stationary stochastic process $\{V_t\}_{t=-\infty}^\infty$ on a probability space with a probability measure $\mathbb{P}$ is said to be *strongly mixing ($\alpha$-mixing)* if a mixing coefficient $\alpha(\tau)$ defined as

$$\alpha(\tau) = \sup_{A\in\mathfrak{M}_{-\infty}^0, B\in\mathfrak{M}_\tau^\infty} |\mathbb{P}(A \cap B) - \mathbb{P}(A)\mathbb{P}(B)|$$

satisfies $\alpha(\tau) \to 0$ as $\tau \to \infty$, where $\mathfrak{M}_s^t$, $-\infty \leq s \leq t \leq \infty$, denotes a $\sigma$-algebra generated by $\{V_s, V_{s+1}, \ldots, V_t\}$. The mixing coefficient $\alpha(\tau)$ quantifies the asymptotic independence between the past and future of the sequence $\{V_t\}_{t=-\infty}^\infty$.

**Assumption 4.2** (Mixing of the process). The stationary process $(V_i = (\tilde{X}_i, \tilde{Y}_i))_{i=1}^\infty$ is strongly mixing with the mixing coefficient $\alpha(\tau) = \mathcal{O}(\tau^{-(2+\delta)})$ for some $\delta > 0$.

We highlight that our strong mixing assumptions apply to the residuals, which is a far less restrictive condition than assuming the original time series itself is strongly mixing. Even when the original time series departs significantly from stationarity, the unobserved noises may still retain stationarity and strong mixing properties. For instance, in a vector auto-regressive model with a time-dependent drift, the noises are drawn from the identical distribution without serial correlation.

Furthermore, the strong mixing property is widely regarded as a relatively weak condition and is commonly met by many time series models, making it a typical assumption in time-series analysis (Cai, 2002). For instance, both linear autoregressive models and the broader class of bilinear models satisfy strong mixing conditions with exponentially decaying mixing coefficients under mild assumptions. Similarly, ARCH processes and nonlinear additive autoregressive models with exogenous variables are recognized for their stationary and strong mixing behavior (Masry & Tjøstheim, 1995; 1997).

Due to stationarity, the conditional CDF of the realized residual does not depend on the index $i$; thus, denote

$$F(b|\tilde{x}) = \mathbb{P}(\tilde{Y}_i \leq b|\tilde{X}_i = \tilde{x}),$$

as the conditional CDF of the random variable $\tilde{Y}_i$ given $\tilde{X}_i = \tilde{x}$. In addition, we introduce the following notations:

- Let $g(\tilde{x})$ be the marginal density of $\tilde{X}_i$ at $\tilde{x}$. (Note that due to stationarity, we can have a common marginal density.)
- Let $g_{1,i}, i \geq 2$ denote the joint density of $\tilde{X}_1$ and $\tilde{X}_i$.

The following assumptions (4.3-4.5) are common in nonparametric statistics, essential for attaining desirable properties such as the consistency of an estimator (Tsybakov, 2009).

**Assumption 4.3** (Smoothness of the conditional CDF and densities). For fixed $\tilde{y} \in \mathbb{R}$ and $\tilde{x} \in \mathbb{R}^w$,

(i) $0 < F(\tilde{y}|\tilde{x}) < 1$.

(ii) $F(\tilde{y}|\tilde{x})$ is twice continuously partially differentiable with respect to $\tilde{x}$.

(iii) $g(\tilde{x}) > 0$ and $g(\cdot)$ is continuous at $\tilde{x}$.

(iv) There exists $M > 0$ such that $|g_{1,i}(u,v) - g(u)g(v)| \leq M$ for all $u, v$ and $i \geq 2$.

Regarding Assumption 4.3, we would like to remark that there is a negative result: without additional assumptions about the distribution, it is impossible to construct finite-length prediction intervals that satisfy conditional coverage (Lei & Wasserman, 2014; Vovk, 2012).

**Assumption 4.4** (Regularity of the kernel function). The kernel $K : \mathbb{R}^w \to \mathbb{R}$ is a nonnegative, bounded, continuous, and compactly supported density function satisfying

(i) $\int_{\mathbb{R}^w} uK(u)du = 0$,

(ii) $\int_{\mathbb{R}^w} uu^\top K(u)du = \mu_2 I$ for some $\mu_2 \in (0,\infty)$,

(iii) $\int_{\mathbb{R}^w} K^2(u)du = \nu_0$ and $\int_{\mathbb{R}^w} uu^\top K^2(u)du = \nu_2 I$ for some $\nu_0, \nu_2 \in (0,\infty)$.

Assumptions 4.4-(i), (ii), and (iii) are standard conditions (Wand & Jones, 1994) that require $K$ to be "symmetric" in a sense that that the weighting scheme relies solely on the distance between the observation and the test point. For example, if $K$ is isometric, i.e., $K(u) = k(\|u\|)$ for some univariate kernel function $k : \mathbb{R} \to \mathbb{R}$, it can satisfy these conditions using widely adopted kernels such as the Epanechnikov kernel.

**Assumption 4.5** (Bandwidth selection). As $n \to \infty$, the bandwidth $h$ satisfies

$$h \to 0, \text{ and } nh^{w(1+2/\delta)} \to \infty.$$

We note that Assumption 4.5 is met when selecting the (theoretically) optimal bandwidth $h^* \sim n^{-1/(w+4)}$, which minimizes the asymptotic mean squared error (AMSE) of the RNW estimator, provided that $\delta > 1/2$.

We prove the following proposition following a similar strategy as (Salha, 2006) by fixing several technical details:

**Proposition 4.6** (Consistency of the RNW estimator). *Under Assumptions 4.2-4.5, given arbitrary $\tilde{x}$ and $\tilde{y}$, as $n \to \infty$,*

$$\widehat{F}(\tilde{y}|\tilde{x}) - F(\tilde{y}|\tilde{x}) = \frac{1}{2}h^2 tr(D_{\tilde{x}}^2(F(\tilde{y}|\tilde{x})))\mu_2 + o_p(h^2) + \mathcal{O}_p((nh^w)^{-1/2}). \tag{14}$$

*where $D_{\tilde{x}}^2 F(\tilde{y}|\tilde{x})$ denotes the Hessian of $F(\tilde{y}|\tilde{x})$ with respect to $\tilde{x}$.*

This proposition implies pointwise convergence in probability of the RNW estimator, and since it is the weighted empirical CDF, this pointwise convergence implies uniform convergence in probability (Tucker, 1967, p.127-128). Consequently, we obtain the consistency of the conditional quantile estimator in equation 10 to the true conditional quantile given as

$$Q_\beta(\tilde{x}) = \inf\{\tilde{y} \in \mathbb{R} : F(\tilde{y}|\tilde{x}) \geq \beta\}.$$

**Corollary 4.7.** *Under Assumptions 4.2-4.5, for every $\beta \in (0,1)$ and $\tilde{x}$, as $n \to \infty$,*

$$\widehat{Q}_\beta(\tilde{x}) \to Q_\beta(\tilde{x}) \text{ in probability.} \tag{15}$$

As a direct consequence of Corollary 4.7, the asymptotic conditional coverage of KOWCPI is guaranteed by the consistency of the quantile estimator used in our sequential algorithm.

**Corollary 4.8** (Asymptotic conditional coverage guarantee). *Under Assumptions 4.2-4.5, for any $\alpha \in (0,1)$, as $n \to \infty$,*

$$\mathbb{P}(Y_t \in \hat{C}_{t-1}^\alpha(X_t)|X_t) \to (1-\alpha) \text{ in probability.} \tag{16}$$

Thus, employing quantile regression using the RNW estimator for prediction residuals derived from the time-series data of continuous random variables, assuming strong mixing of these residuals, KOWCPI can achieve approximate conditional coverage with a sufficient number of residuals utilized.

To further specify the rate of convergence, define the *discrete gap*

$$\Delta(\tilde{X}) := \sup_{\beta \in [0,1]} |\widehat{F}(\widehat{Q}_\beta(\tilde{X})|\tilde{X}) - \beta| = \max_{i=1,\dots,n} \widehat{W}_i(\tilde{X}), \tag{17}$$

introduced by the quantile estimator being the generalized inverse distribution function.

**Theorem 4.9** (Conditional coverage gap). *Under Assumptions 4.2-4.5, for any $\alpha \in (0,1)$ and $x_t$, as $n \to \infty$,*

$$\left|\mathbb{P}\left(Y_t \in \widehat{C}_{t-1}^\alpha(x_t) \mid X_t = x_t\right) - (1-\alpha)\right| \leq \mathcal{O}_p(h^2 + (nh^w)^{-1/2}) + 2\Delta(\tilde{x}_{n+1}), \tag{18}$$

*where $\tilde{x}_{n+1}$ is the realization of $\tilde{X}_{n+1}$ given $X_t = x_t$.*

Given that the adjustment weights $p_i(\tilde{x})$ uniformly concentrate to $1/n$ (Steikert, 2014), one can see that the conditional coverage gap tends to zero, although its precise rate remains an open question.

## 5 EXPERIMENTS

In this section, we compare the performance of KOWCPI against state-of-the-art conformal prediction baselines using real time-series data. Additional experimental results, not included in this section, using both real and synthetic data, are provided in Appendices C and D. We aim to show that KOWCPI can consistently reach valid coverage with the narrowest prediction intervals.

*Dataset.* We consider three real time series from different domains. The first ELEC2 data set (electric) (Harries, 1999) tracks electricity usage and pricing in the states of New South Wales and Victoria in Australia for every 30 minutes over a 2.5-year period in 1996–1999. The second renewable energy data (solar) (Zhang et al., 2021) are from the National Solar Radiation Database and contain hourly solar radiation data (measured in GHI) from Atlanta in 2018. The third wind speed data (wind) (Zhu et al., 2021) are collected at wind farms operated by MISO in the US. The wind speed record was updated every 15 minutes over a one-week period in September 2020.

*Baselines.* We consider Sequential Predictive Conformal Inference (SPCI) (Xu & Xie, 2023b), Ensemble Prediction Interval (EnbPI) (Xu & Xie, 2023a), Adaptive Conformal Inference (ACI) (Gibbs & Candès, 2021), Aggregated ACI (AgACI) (Zaffran et al., 2022), Fully Adaptive Conformal Inference (FACI) (Gibbs & Candès, 2024), Scale-Free Online Gradient Descent (SF-OGD) (Orabona & Pál, 2018; Bhatnagar et al., 2023), Strongly Adaptive Online Conformal Prediction (SAOCP) (Bhatnagar et al., 2023), and vanilla Split Conformal Prediction (SCP) (Vovk et al., 2005). Additionally, we included a comparison where weights were derived from the original Nadaraya-Watson estimator

Table 1: Empirical marginal coverage and average width across three real time-series datasets by different methods. The target coverage is $1 - \alpha = 0.9$. The values in the bracket are standard deviation across five independent trials.

| | Electric | | Wind | | Solar | |
|---|---|---|---|---|---|---|
| | Coverage | Width | Coverage | Width | Coverage | Width |
| KOWCPI | 0.90 (2.3e-3) | 0.22 (1.5e-3) | 0.91 (2.8e-3) | 2.41 (3.2e-2) | 0.90 (1.2e-3) | 48.8 (9.4e-1) |
| Plain NW | 0.89 (1.7e-3) | 0.31 (2.2e-3) | 0.95 (7.4e-3) | 3.58 (1.0e-1) | 0.41 (2.7e-3) | 20.1 (1.8e+0) |
| SPCI | 0.90 (1.1e-3) | 0.29 (1.9e-3) | 0.94 (1.0e-2) | 2.61 (2.1e-2) | 0.92 (1.7e-3) | 84.2 (1.7e+0) |
| EnbPI | 0.93 (3.4e-3) | 0.36 (2.7e-3) | 0.92 (2.3e-3) | 5.25 (4.3e-2) | 0.87 (1.1e-3) | 106.0 (2.3e+0) |
| ACI | 0.89 (0.0e-0) | 0.32 (2.0e-3) | 0.88 (0.0e-0) | 8.26 (2.8e-2) | 0.89 (1.0e-3) | 143.9 (2.3e-1) |
| FACI | 0.89 (2.5e-3) | 0.28 (1.2e-3) | 0.91 (3.2e-3) | 7.77 (1.7e-1) | 0.89 (0.0e-0) | 141.9 (6.4e-1) |
| AgACI | 0.91 (3.1e-3) | 0.30 (2.3e-3) | 0.88 (1.1e-2) | 7.54 (1.2e-1) | 0.90 (2.4e-3) | 144.6 (1.4e+0) |
| SF-OGD | 0.79 (5.8e-4) | 0.25 (1.0e-3) | 0.11 (2.6e-3) | 0.29 (7.0e-4) | 0.00 (0.0e-0) | 0.50 (0.0e-0) |
| SAOCP | 0.93 (6.1e-3) | 0.33 (2.4e-3) | 0.76 (1.1e-2) | 4.00 (4.5e-2) | 0.64 (1.9e-3) | 33.5 (7.3e-2) |
| SCP | 0.87 (2.8e-3) | 0.30 (5.9e-4) | 0.86 (3.2e-3) | 8.20 (1.5e-2) | 0.89 (1.0e-3) | 142.0 (3.8e-1) |

(Plain NW). For the implementation of ACI-related methods, we utilized the R package AdaptiveConformal (`https://github.com/herbps10/AdaptiveConformal`). For SPCI and EnbPI, we used the Python code from `https://github.com/hamrel-cxu/SPCI-code`.

*Setup and evaluation metrics.* In all comparisons, we use the random forest as the base point predictor with the number of trees $= 10$. Every dataset is split in a 7:1:2 ratio for training the point predictor, tuning the window length $w$ and bandwidth $h$, and constructing prediction intervals, respectively. The window length for each dataset is fixed and determined through cross-validation, while the bandwidth is selected by minimizing the nonparametric AIC, as detailed in equation 11.

Besides examining marginal coverage and widths of prediction intervals on test data, we also focus on *rolling coverage*, which is helpful in showing approximate conditional coverage at specific time indices. Given a rolling window size $m > 0$, rolling coverage $\widehat{\mathrm{RC}}_t$ at time $t$ is defined as $\widehat{\mathrm{RC}}_t = \frac{1}{m} \sum_{i=1}^{m} \mathbb{1}\{Y_{t-i+1} \in \widehat{C}_{t-i}^{\alpha}(X_{t-i+1})\}$.

*Results.* The empirical marginal coverage and width results for all methods are summarized in Table 1. The results indicate that KOWCPI consistently achieves the 90% target coverage and maintains the smallest average width compared to the alternative state-of-the-art methods. While all methods, except SF-OGD, SAOCF, and Plain NW nearly achieve marginal coverage under target $1 - \alpha = 0.9$, KOWCPI produces the narrowest average width on all datasets. In terms of rolling results, we show in Figures 2a and 2c that the coverage of KOWCPI intervals consistently centers around 90% throughout the entire test phase. Additionally, Figure 2b shows that KOWCPI intervals are also significantly narrower with a smaller variance than the baselines.

Lastly, Figure 2d depicts the weights $\widehat{W}$ (in log scale) assigned by the RNW estimator at the first time index of the test data. Notably, the most recent set of non-conformity scores (in terms of time indices) is assigned the heaviest weights, which aligns intuitively as these are the most similar to the first test datum in terms of temporal proximity. We believe that this heavy weighting of recent residuals contributes significantly to KOWCPI's performance. Datasets where KOWCPI demonstrates significant superiority, such as the Solar dataset, typically exhibit active volatility changes. In these cases, KOWCPI adapts quickly to changing conditions by leveraging the high weights assigned to recent residuals. For instance, in Figure A.3, which visualizes the performance of KOWCPI, SPCI, and ACI on the Solar dataset, KOWCPI dynamically adjusts its interval widths to reflect whether it is in a high or low-volatility region. This adaptive behavior allows KOWCPI to avoid over-coverage and maintain narrower average widths compared to methods like SPCI and ACI, which produce intervals with relatively constant widths across all regions. At the same time, we acknowledge that such fast-adapting behavior, avoiding conservative intervals, can occasionally lead to brief coverage failures in some regions due to the aggressive adaptation to rapidly changing conditions.

In Appendix C.1, we show additional comparisons of KOWCPI against the baselines on the other two datasets in terms of rolling results. See Appendices C.2 and D for additional experimental results

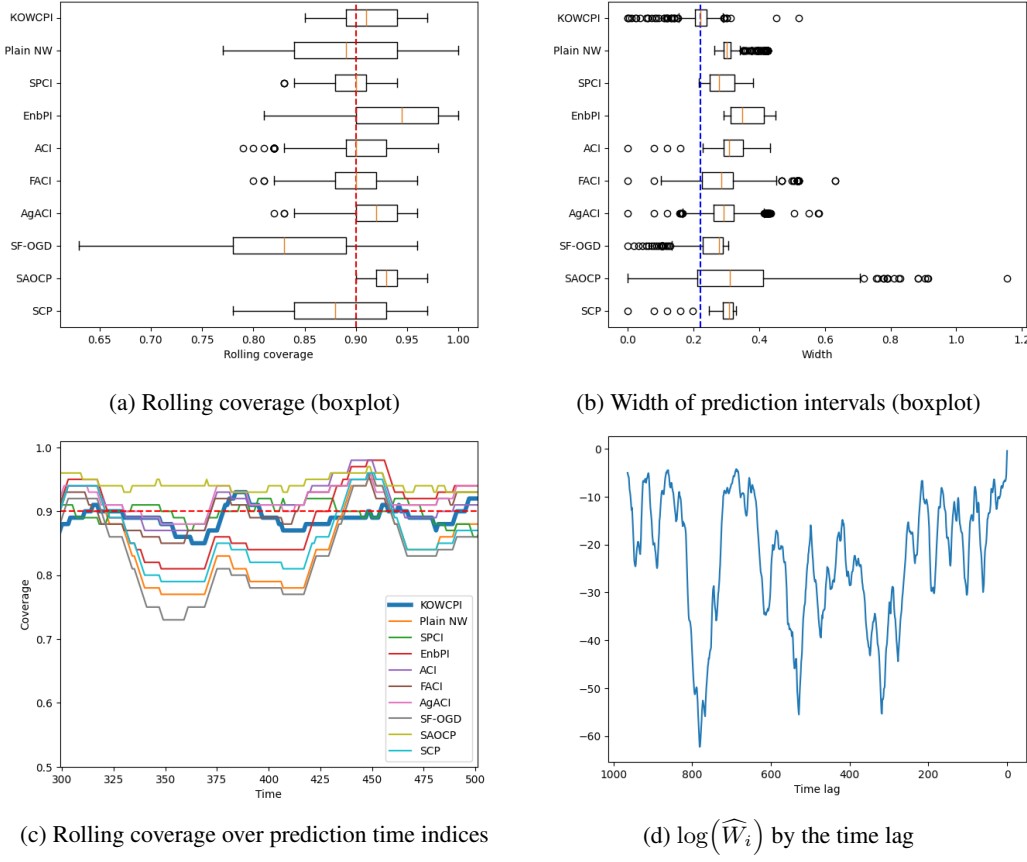

(a) Rolling coverage (boxplot)

(b) Width of prediction intervals (boxplot)

(c) Rolling coverage over prediction time indices

(d) $\log\left(\widehat{W_i}\right)$ by the time lag

Figure 2: Comparison of empirical rolling coverage and width on the electric dataset by different methods in (a) rolling coverage; (b) widths of intervals, (c) rolling coverage over time, and (d) an instance of computed final weights. The target coverage is 90%. In (a), the red dotted line is the target coverage and in (b), the blue dotted line is the median width of KOWCPI.

using a wider variety of real and synthetic datasets, where we consistently observe the coverage validity of KOWCPI while yielding the shortest intervals on average.

## 6 CONCLUSION

In this paper, we introduced KOWCPI, a method to sequentially construct prediction intervals for time-series data. By incorporating the classical Reweighted Nadaraya-Watson estimator into the weighted conformal prediction framework, KOWCPI effectively adapts to the dependent structure of time-series data by utilizing data-driven adaptive weights. Our theoretical contributions include providing theoretical guarantees for the asymptotic conditional coverage of KOWCPI under strong mixing conditions and bounding the marginal and conditional coverage gaps. Empirical validation on real-world time-series datasets demonstrated the effectiveness of KOWCPI compared to state-of-the-art methods, achieving narrower prediction intervals without compromising empirical coverage.

Future work could explore *adaptive window selection*, where the size of the non-conformity score batch is adjusted dynamically to capture shifts in the underlying distribution. A preliminary implementation of this approach is discussed in Appendix E, showcasing its potential to improve flexibility and adaptability in practice. Additionally, the natural compatibility of kernel regression with multivariate data can be leveraged to expand the utility of KOWCPI for multivariate time-series data, as detailed in Appendix A. There is also potential for improving theoretical guarantees and practical performance by designing alternative non-conformity scores.

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

# A   MULTIVARIATE TIME SERIES

In the main text, our discussion has centered on cases where the response variables $Y_t$ are scalars. Here, we explore the natural extension of our methodology to handle scenarios with multivariate responses. This extension requires defining multivariate quantiles, introducing a multivariate version of the RNW estimator for estimating these quantiles (Salha, 2006), and adapting our KOWCPI method for multivariate responses.

**Multivariate conditional quantiles**   Consider a strongly mixing stationary process $((\tilde{X}_i, \tilde{Y}_i))_{i=1}^\infty$, which is a realization of random variable $(\tilde{X}, \tilde{Y}) \in \mathbb{R}^p \times \mathbb{R}^s$. Following Abdous & Theodorescu (1992), we first define a pseudo-norm function $\|\cdot\|_{2,\alpha} : \mathbb{R}^s \to \mathbb{R}$ for $\alpha \in (0,1)$ as

$$\|v\|_{2,\alpha} = \left\| \left( \frac{|v_1| + (2\alpha - 1)}{2}, \ldots, \frac{|v_s| + (2\alpha - 1)}{2} \right) \right\|_2,$$

for $v \in \mathbb{R}^s$, where $\|\cdot\|_2$ is the Euclidean norm on $\mathbb{R}^s$. Let

$$H_\alpha(\theta, \tilde{x}) := \mathbb{E}[\|\tilde{Y} - \theta\|_{2,\alpha} - \|\tilde{Y}\|_{2,\alpha} \mid \tilde{X} = \tilde{x}].$$

**Definition A.1** (Multivariate conditional quantile (Abdous & Theodorescu, 1992)). Define a multivariate conditional $\alpha$-quantile $\theta_\alpha(\tilde{x})$ for $\alpha \in (0,1)$ as

$$\theta_\alpha(\tilde{x}) = \operatorname*{argmin}_{\theta \in \mathbb{R}^s} H_\alpha(\theta, \tilde{x}). \tag{A.1}$$

*Remark* A.2 (Compatibility with univariate quantile function). For a scalar $\tilde{Y} \in \mathbb{R}$, its conditional quantile given $\tilde{X} = \tilde{x}$ is

$$
\begin{aligned}
\theta_\alpha(\tilde{x}) &= \operatorname*{argmin}_{\theta \in \mathbb{R}} \mathbb{E}\left[ \|\tilde{Y} - \theta\|_{2,\alpha} \mid \tilde{X} = \tilde{x} \right] \\
&= \operatorname*{argmin}_{\theta \in \mathbb{R}} \mathbb{E}\left[ |\tilde{Y} - \theta| + (2\alpha - 1)(\tilde{Y} - \theta) \mid \tilde{X} = \tilde{x} \right] \\
&= \operatorname*{argmin}_{\theta \in \mathbb{R}} \mathbb{E}((\tilde{Y} - \theta)(\alpha - \mathbb{1}(\tilde{Y} \le \theta)) \mid \tilde{X} = \tilde{x}) \\
&= Q_\alpha(\tilde{x}),
\end{aligned}
$$

for any $\alpha \in (0,1)$. Thus, Definition A.1 is consistent with the univariate case.

**Multivariate RNW estimator**   Following the definition in equation 6, we obtain the RNW estimator for multivariate responses as

$$\widehat{F}(\tilde{y}|\tilde{x}) = \frac{\sum_{i=1}^n W(\frac{\tilde{X}_i - \tilde{x}}{h}) \mathbb{1}(\tilde{Y}_i \le \tilde{y})}{\sum_{i=1}^n W(\frac{\tilde{X}_i - \tilde{x}}{h})}, \tag{A.2}$$

where, according to equation 7 and equation 8,

$$W(u) = \frac{K(u)}{1 + \lambda u_1 K(u)},$$

for $u = (u_1, \ldots, u_p)^\top$. Now, let $W_h(u) = h^{-p} W(h^{-1} u)$, and define an estimator for $H_\alpha(\theta, x)$ as

$$\widehat{H}_\alpha(\theta, \tilde{x}) := \int_{\mathbb{R}^s} (\|\tilde{y} - \theta\|_{2,\alpha} - \|\tilde{y}\|_{2,\alpha}) \widehat{F}(d\tilde{y}|\tilde{x}) = \frac{\sum_{i=1}^n W_h(\tilde{X}_i - \tilde{x}) \left( \|\tilde{Y}_i - \theta\|_{2,\alpha} - \|\tilde{Y}_i\|_{2,\alpha} \right)}{\sum_{i=1}^n W_h(\tilde{X}_i - \tilde{x})},$$

and consequently the RNW conditional quantile estimator $\widehat{\theta}_\alpha$ as

$$\widehat{\theta}_\alpha(\tilde{x}) := \operatorname*{argmin}_{\theta \in \mathbb{R}^p} \widehat{H}_\alpha(\theta, \tilde{x}) = \operatorname*{argmin}_{\theta \in \mathbb{R}^p} \sum_{i=1}^n W_h(\tilde{X}_i - \tilde{x})(\|\tilde{Y}_i - \theta\|_{2,\alpha} - \|\tilde{Y}_i\|_{2,\alpha}). \tag{A.3}$$

**Multivariate `KOWCPI`**   Suppose we are sequentially observing $(X_t, Y_t) \in \mathbb{R}^d \times \mathbb{R}^s$, $t = 1, 2, \ldots$. Based on the construction of the multivariate version of the RNW estimator, we can extend our `KOWCPI` approach to multivariate responses in the same manner as described in Algorithm 1, with multivariate residuals

$$\hat{\varepsilon}_t = Y_t - \hat{f}(X_t) \in \mathbb{R}^s,$$

as non-conformity scores. This adaptation allows for the application of our methodology to a broader range of data scenarios involving dependent data with multivariate response variables, which were similarly studied in (Xu et al., 2024; Sun & Yu, 2024; Stankevičiūtė et al., 2021).

## B   PROOFS

The following lemma is adapted from the proof of Lemma 1 of Tibshirani et al. (2019); however, we do not assume exchangeability.

**Lemma B.1** (Weights on quantile for non-exchangeable data). *Given a sequence of random variables* $\{V_1, \ldots, V_{n+1}\}$ *with joint density* $\mathcal{P}$ *and a sequence of observations* $\{v_1, \ldots, v_{n+1}\}$. *Define the event*

$$E = \{\{V_1, \ldots, V_{n+1}\} = \{v_1, \ldots, v_{n+1}\}\}.$$

*Then we have for* $i = 1, \ldots, n + 1$,

$$\mathbb{P}\{V_{n+1} = v_i | E\} = \frac{\sum_{\sigma : \sigma(n+1) = i} \mathcal{P}(v_{\sigma(1)}, \ldots, v_{\sigma(n+1)})}{\sum_{\sigma} \mathcal{P}(v_{\sigma(1)}, \ldots, v_{\sigma(n+1)})} \in [0, 1].$$

Note that when the residuals are exchangeable, $W_i^* = 1/(n+1)$, as also observed in Tibshirani et al. (2019). Now we prove Proposition 4.1.

*Proof of Proposition 4.1.*   The proof assumes that $\tilde{Y}_i$, for $i = 1, \ldots, n+1$, are almost surely distinct. However, the proof remains valid, albeit with more complex notations involving multisets, if this is not the case. Denote by $\text{Quantile}_\beta(\mathbb{Q})$ the $\beta$-quantile of the distribution $\mathbb{Q}$ on $\mathbb{R}$, and by $\delta_a$ the point mass distribution at $a \in \mathbb{R}$. Define the event $E = \{\{\tilde{Y}_1, \ldots, \tilde{Y}_{n+1}\} = \{v_1, \ldots, v_{n+1}\}\}$. Then, by the tower property, we have

$$
\begin{aligned}
&\mathbb{P}(Y_{T+1} \in \widehat{C}_T^\alpha(X_{T+1})) \\
&= \mathbb{E}(\mathbb{P}(Y_{T+1} \in \widehat{C}_T^\alpha(X_{T+1}) \mid E)) \\
&= \mathbb{E}\left[\mathbb{P}\left(\text{Quantile}_{\beta^*}\left(\sum_{i=1}^n \widehat{W}_i \delta_{v_i}\right) \le \tilde{Y}_{n+1} \le \text{Quantile}_{1-\alpha+\beta^*}\left(\sum_{i=1}^n \widehat{W}_i \delta_{v_i}\right)\middle| E\right)\right] \\
&= \mathbb{E}\left[\mathbb{P}_{V \sim P^{W^*}}\left(\text{Quantile}_{\beta^*}\left(\sum_{i=1}^n \widehat{W}_i \delta_{v_i}\right) \le V \le \text{Quantile}_{1-\alpha+\beta^*}\left(\sum_{i=1}^n \widehat{W}_i \delta_{v_i}\right)\right)\right],
\end{aligned}
$$

where $P^{W^*} = \sum_{i=1}^{n+1} W_i^* \delta_{v_i}$, and in the last line, we have used the result from Lemma B.1,

$$\tilde{Y}_{n+1} | E \sim P^{W^*}.$$

Denote the weighted empirical distributions based on $\widehat{W} = \widehat{W}(\tilde{X}_{n+1})$ as

$$P^{\widehat{W}} = \sum_{i=1}^n \widehat{W}_i \delta_{v_i}.$$

This gives the marginal coverage gap as

$$
\left| \mathbb{P}(Y_{n+1} \in \widehat{C}_n^\alpha(X_{n+1})) - (1-\alpha) \right|
$$

$$
\leq \mathbb{E}\left[ \left| \mathbb{P}_{V \sim P^{W^*}}\left( \text{Quantile}_{\beta^*}\left( \sum_{i=1}^n \widehat{W}_i \delta_{v_i} \right) \leq V \leq \text{Quantile}_{1-\alpha+\beta^*}\left( \sum_{i=1}^n \widehat{W}_i \delta_{v_i} \right) \right) \right. \right.
$$

$$
\left. \left. - \mathbb{P}_{V \sim P^{\widehat{W}}}\left( \text{Quantile}_{\beta^*}\left( \sum_{i=1}^n \widehat{W}_i \delta_{v_i} \right) \leq V \leq \text{Quantile}_{1-\alpha+\beta^*}\left( \sum_{i=1}^n \widehat{W}_i \delta_{v_i} \right) \right) \right| \right]
$$

$$
+ \mathbb{E}\left| \mathbb{P}_{V \sim P^{\widehat{W}}}\left( \text{Quantile}_{\beta^*}\left( \sum_{i=1}^n \widehat{W}_i \delta_{v_i} \right) \leq V \leq \text{Quantile}_{1-\alpha+\beta^*}\left( \sum_{i=1}^n \widehat{W}_i \delta_{v_i} \right) \right) - (1-\alpha) \right|
$$

$$
\leq \mathbb{E}[d_{\mathrm{TV}}(P^{W^*}, P^{\widehat{W}})]
$$

$$
+ \mathbb{E}\left| \widehat{F}\left( \widehat{Q}_{1-\alpha+\beta^*}(\tilde{X}_{n+1}) \Big| \tilde{X}_{n+1} \right) - (1-\alpha+\beta^*) \right| + \mathbb{E}\left| \widehat{F}\left( \widehat{Q}_{\beta^*}(\tilde{X}_{n+1}) \Big| \tilde{X}_{n+1} \right) - \beta^* \right|
$$

$$
\leq \frac{1}{2}\mathbb{E}\|(W^*)_{1:n} - \widehat{W}\|_1 + \frac{1}{2}\mathbb{E}W_{n+1}^* + 2\mathbb{E}\max_{i=1,\dots,n} \widehat{W}_i(\tilde{X}_{n+1}),
$$

where we denote by $d_{\mathrm{TV}}(\cdot, \cdot)$ the total variation distance between probability measures, and the second inequality is due to the definition of the total variation distance. □

### B.1 PROOF OF ASYMPTOTIC CONDITIONAL COVERAGE OF KOWCPI (COROLLARY 4.8 AND THEOREM 4.9)

In deriving the asymptotic conditional coverage property of KOWCPI, the consistency of the RNW estimator plays a crucial role. Therefore, we first introduce the proof of Proposition 4.6, which discusses the consistency of the CDF estimator. Corollary 4.7, which states the consistency of the quantile estimator, is a natural consequence of Proposition 4.6 and leads us to the proof for our main results, Corollary 4.8 and Theorem 4.9. Proof of Proposition 4.6 adopts the similar strategy as Salha (2006) and Cai (2002).

To prove Proposition 4.6, it is essential to first understand the nature of the adjustment weight $p_i(\tilde{x})$. Thus, Lemma 3.1 is not only crucial for the practical implementation of the RNW estimator but also indispensable in the proof process of Proposition 4.6.

*Proof of Lemma 3.1.* For display purposes, denote $[X]_1$ as $X_1$. By equation 5, we have that

$$
\sum_{i=1}^n p_i(\tilde{x})(\tilde{X}_{i1} - \tilde{x}_1)K_h(\tilde{X}_i - \tilde{x}) = 0. \tag{A.4}
$$

Let

$$
\mathcal{L}(\lambda_1, \lambda_2, p_1(\tilde{x}), \dots, p_n(\tilde{x}))
$$

$$
= \sum_{i=1}^n \log p_i(\tilde{x}) + \lambda_1\left( 1 - \sum_{i=1}^n p_i(\tilde{x}) \right) + \lambda_2 \sum_{i=1}^n p_i(\tilde{x})(\tilde{X}_{i1} - \tilde{x}_1)K_h(\tilde{X}_i - \tilde{x}),
$$

where $\lambda_1, \lambda_2 \in \mathbb{R}$ are the Lagrange multipliers. From $\partial\mathcal{L}/\partial p_i(\tilde{x}) = 0$ for $i = 1, \dots, n$, we get

$$
p_i^{-1}(\tilde{x}) - \lambda_1 + \lambda_2(\tilde{X}_{i1} - \tilde{x}_1)K_h(\tilde{X}_i - \tilde{x}) = 0,
$$

Since $p_i(\tilde{x})$'s sum up to 1 as in equation 4, letting $\lambda = -\lambda_2/\lambda_1$, we have

$$
p_i(\tilde{x}) = \frac{[1 + \lambda(\tilde{X}_{i1} - \tilde{x}_1)K_h(\tilde{X}_i - \tilde{x})]^{-1}}{\sum_{j=1}^n [1 + \lambda(\tilde{X}_{j1} - \tilde{x}_1)K_h(\tilde{X}_j - \tilde{x})]^{-1}}.
$$

Using equation 4 again with equation A.4, this gives

$$
\sum_{j=1}^n \left[ 1 + \lambda(\tilde{X}_{j1} - \tilde{x}_1)K_h(\tilde{X}_j - \tilde{x}) \right]^{-1} = n\left( \sum_{i=1}^n p_i(\tilde{x})[1 + \lambda(\tilde{X}_{i1} - \tilde{x}_1)K_h(\tilde{X}_i - \tilde{x})] \right)^{-1} = n,
$$

and therefore equation 8 holds. With equation 5, this gives

$$0 = \sum_{i=1}^{n} \frac{(\tilde{X}_{i1} - \tilde{x}_1)K_h(\tilde{X}_i - \tilde{x})}{1 + \lambda(\tilde{X}_{i1} - \tilde{x}_1)K_h(\tilde{X}_i - \tilde{x})} = -\left. \frac{\partial L(\gamma; \tilde{x})}{\partial \gamma} \right|_\lambda.$$

Note that $\frac{\partial^2 L(\gamma; \tilde{x})}{\partial \gamma^2} \geq 0$, implying that $L(\cdot; \tilde{x})$ is indeed a convex function. □

**Lemma B.2.** *Under the assumptions of Proposition 4.6, define*

$$c(\tilde{x}) = \frac{\frac{\partial g(\tilde{x})}{\partial \tilde{x}_1} \mu_2}{g(\tilde{x})\nu_2}.$$

*Then,*

$$\lambda = h^w \cdot c(\tilde{x})(1 + o_p(1)) = \mathcal{O}_p(h^w). \tag{A.5}$$

*Proof.* Let

$$S_i = (\tilde{X}_{i1} - \tilde{x}_1)K_h(\tilde{X}_i - \tilde{x}).$$

Then, by Assumption 4.4, $S_i$ is bounded above by some constant $C_1$. Let

$$\overline{S^k} = \frac{1}{n}\sum_{i=1}^{n}(S_i)^k,$$

for $k = 1, 2$. Then, from equation A.4, we have

$$0 = \frac{1}{n}\sum_{i=1}^{n}(1 + \lambda S_i)^{-1} S_i \geq |\lambda| \left| \frac{1}{n}\sum_{i=1}^{n} S_i^2(1 + \lambda S_i)^{-1} \right| - \left| \overline{S^1} \right| \geq |\lambda|(1 + C_1|\lambda|)^{-1}\overline{S^2} - \left| \overline{S^1} \right|,$$

which gives

$$|\lambda| \leq \frac{\left| \overline{S^1} \right|}{\overline{S^2} - C_1\left| \overline{S^1} \right|}.$$

Using the Taylor expansion (Wand & Jones, 1994), we obtain that

$$\mathbb{E}\overline{S^1} = \int_{\mathbb{R}^w} (u_1 - \tilde{x}_1)K_h(u - \tilde{x})g(u)du$$

$$= h \int_{\mathbb{R}^w} u_1 K(u)g(\tilde{x} + hu)du$$

$$= h \int_{\mathbb{R}^w} u_1 K(u)\left( g(\tilde{x}) + h\sum_{j=1}^{w} u_j \frac{\partial g(\tilde{x})}{\partial \tilde{x}_j} \right)du + o(h^2)$$

$$= h^2 \left\{ \frac{\partial g(\tilde{x})}{\partial \tilde{x}_1}\mu_2 + o_p(1) \right\},$$

where the last equation comes from Assumptions 4.4-(i) and (ii). With a similar argument, we can derive that

$$\mathbb{E}\overline{S^2} = \int_{\mathbb{R}^w} (u_1 - \tilde{x}_1)^2 K_h^2(u - \tilde{x})g(u)du = h^{-w+2}\left\{ g(\tilde{x})\nu_2 + o_p(h) \right\},$$

using Assumption 4.4-(iii). Therefore, we obtain equation A.5. □

Decomposing $\widehat{F}(\tilde{y}|\tilde{x}) - F(\tilde{y}|\tilde{x})$ in bias and variance terms, we get

$$\widehat{F}(\tilde{y}|\tilde{x}) - F(\tilde{y}|\tilde{x})$$

$$= \frac{\sum_{i=1}^{n} p_i(\tilde{x})K_h(\tilde{X}_i - \tilde{x})\{\mathbb{1}(\tilde{Y}_i < \tilde{y}) - F(\tilde{y}|\tilde{x})\}}{\sum_{i=1}^{n} p_i(\tilde{x})K_h(\tilde{X}_i - \tilde{x})} \tag{A.6}$$

$$= \frac{\sum_{i=1}^{n} p_i(\tilde{x})K_h(\tilde{X}_i - \tilde{x})\delta_i + \sum_{i=1}^{n} p_i(\tilde{x})K_h(\tilde{X}_i - \tilde{x})\{F(\tilde{y}|\tilde{X}_i) - F(\tilde{y}|\tilde{x})\}}{\sum_{i=1}^{n} p_i(\tilde{x})K_h(\tilde{X}_i - \tilde{x})},$$

where $\delta_i = \mathbb{1}(\tilde{Y}_t \leq \tilde{y}) - F(\tilde{y}|\tilde{X}_i)$. Note that $\mathbb{E}[\delta_i] = 0$ due to the tower property. Now, let

$$b_i(\tilde{x}) = b_i(\tilde{X}_i, \tilde{x}) := \left[ 1 + h^w \cdot c(\tilde{x})(\tilde{X}_{i1} - \tilde{x}_1)K_h(\tilde{X}_i - \tilde{x}) \right]^{-1}.$$

Then, by Lemma B.2, we have that

$$p_i(\tilde{x}) = n^{-1}b_i(\tilde{x})(1 + o_p(1)). \tag{A.7}$$

Define the approximations for the terms in the decomposition presented in equation A.6:

$$J_1 = n^{-1/2}h^{w/2} \sum_{i=1}^n b_i(\tilde{x})\delta_i K_h(\tilde{X}_i - \tilde{x}),$$

$$J_2 = n^{-1} \sum_{i=1}^n \left\{ F(\tilde{y}|\tilde{X}_i) - F(\tilde{y}|\tilde{x}) \right\} b_i(\tilde{x})K_h(\tilde{X}_i - \tilde{x}),$$

$$J_3 = n^{-1} \sum_{i=1}^n b_i(\tilde{x})K_h(\tilde{X}_i - \tilde{x}),$$

so that

$$\widehat{F}(\tilde{y}|\tilde{x}) - F(\tilde{y}|\tilde{x}) = \{(nh^w)^{-1/2}J_1 + J_2\}J_3^{-1}\{1 + o_p(1)\}. \tag{A.8}$$

Therefore, we will derive Proposition 4.6 by controlling the terms $J_1$, $J_2$ and $J_3$.

**Lemma B.3.** *Under the assumptions of Proposition 4.6,*

$$J_1 = \mathcal{O}_p(1). \tag{A.9}$$

*Proof.* Let

$$\xi_i = h^{w/2}b_i(\tilde{x})\delta_i K_h(\tilde{X}_i - \tilde{x}),$$

so that $J_1 = n^{-1/2} \sum_{i=1}^n \xi_i$. Since $\mathbb{E}(\delta_i|\tilde{X}_i) = 0$, we have that $\mathbb{E}(\xi_i) = \mathbb{E}(\mathbb{E}(\xi_i|\tilde{X}_i)) = 0$, and thus

$$\mathbb{E}J_1 = 0. \tag{A.10}$$

Also, due to the stationarity of $\tilde{X}_i$, we have that

$$\mathrm{Var}(J_1) = \mathbb{E}\xi_i^2 + \sum_{i=2}^n \left( 1 - \frac{i-1}{n} \right) \mathrm{Cov}(\xi_1, \xi_i). \tag{A.11}$$

By Assumption 4.4, we have that $\lim_{n\to\infty} b_i(\tilde{x}) = 1$, which gives $\mathbb{E}(b_i) = 1 + o_p(1)$. Therefore, through expansion, we have

$$\begin{aligned} \mathbb{E}\xi_i^2 &= h^w \mathbb{E}\left[ \mathbb{E}\left[ b_i^2(\tilde{x})K_h^2(\tilde{X}_i - \tilde{x})\delta_i^2 \mid \tilde{X}_i \right] \right] \\ &= h^w \mathbb{E}\left[ b_i^2(\tilde{x})K_h^2(\tilde{X}_i - \tilde{x})F(\tilde{y}|\tilde{X}_i)(1 - F(\tilde{y}|\tilde{X}_i)) \right] \\ &= \left[ (K^2)_h * \{b_i^2(\cdot, \tilde{x})F(\tilde{y}|\cdot)(1 - F(\tilde{y}|\cdot))g(\cdot)\} \right](\tilde{x}) \\ &= \nu_0 F(\tilde{y}|\tilde{x})(1 - F(\tilde{y}|\tilde{x}))g(\tilde{x}) + o_p(1), \end{aligned}$$

where $*$ in the third line is the convolution operator. To control the second term in the right-hand side of equation A.11, we borrow the idea of Masry (1986). Choose $d_n = \mathcal{O}(h^{-\frac{w}{1+\delta/2}})$ and decompose

$$\sum_{i=2}^n \left( 1 - \frac{i-1}{n} \right) \mathrm{Cov}(\xi_1, \xi_i) = \sum_{i=2}^{d_n} \left( 1 - \frac{i-1}{n} \right) \mathrm{Cov}(\xi_1, \xi_i) + \sum_{i=d_n+1}^n \left( 1 - \frac{i-1}{n} \right) \mathrm{Cov}(\xi_1, \xi_i).$$

We have that $|b_i(\tilde{x})\delta_i| \leq C_2$ for some constant $C_2$. By Assumption 4.3-(iv), we obtain

$$\begin{aligned} |\mathrm{Cov}(\xi_1, \xi_i)| &= \left| \int_{\mathbb{R}^w} \int_{\mathbb{R}^w} \xi_1 \xi_i g_{1,i}(u, v) du dv - \int_{\mathbb{R}^w} \xi_1 g(u) du \int_{\mathbb{R}^w} \xi_i g(v) dv \right| \\ &\leq C_2^2 h^w \int_{\mathbb{R}^w} \int_{\mathbb{R}^w} K(u)K(v)|g_{1,i}(\tilde{x} - hu, \tilde{x} - hv) - g(\tilde{x} - hu)g(\tilde{x} - hv)| du dv \\ &\leq C_2^2 M h^w, \end{aligned}$$

so that

$$\sum_{i=2}^{d_n} \left(1 - \frac{i-1}{n}\right) \mathrm{Cov}(\xi_1, \xi_i) = \mathcal{O}_p(d_n h^w) = o_p(1).$$

By Assumption 4.4, we have $\|(\tilde{X}_i - \tilde{x})K_h(\tilde{X}_i - \tilde{x})\| \leq C_3$, so that $|\xi_i| \leq Ch^{-w/2}$. Then, by Theorem 17.2.1 of Ibragimov et al. (1971), we have that

$$|\mathrm{Cov}(\xi_1, \xi_i)| \leq Ch^{-w}\alpha(i-1).$$

Thus, we get

$$\sum_{i=d_n+1}^{n} \left(1 - \frac{i-1}{n}\right) \mathrm{Cov}(\xi_1, \xi_i) \leq Ch^{-w} \sum_{i \geq d_n} \alpha(i) \leq Ch^{-w} d_n^{-1-\delta} = o(1).$$

Therefore, we obtain

$$J_1 = \mathbb{E}J_1 + \mathcal{O}_p\left(\sqrt{\mathrm{Var}(J_1)}\right) = \mathcal{O}_p(1). \tag{A.12}$$

$\square$

**Lemma B.4.** *Under the assumptions of Proposition 4.6,*

$$J_2 = \frac{1}{2}h^2\mu_2 tr(D_{\tilde{x}}^2 F(\tilde{y}|\tilde{x}))g(\tilde{x}) + o_p(h^2), \tag{A.13}$$

$$J_3 = g(\tilde{x}) + o_p(1). \tag{A.14}$$

*Proof.* By Assumption 1, equation 4 and equation 5, we have through expansion that

$$J_2 = (2n)^{-1} \sum_{i=1}^{n} (\tilde{X}_i - \tilde{x})^\top (D_{\tilde{x}}^2 F(\tilde{y}|\tilde{x}))(\tilde{X}_i - \tilde{x})b_i(\tilde{x})K_h(\tilde{X}_i - \tilde{x}) + o_p(h^2).$$

Since

$$\mathbb{E}[(\tilde{X}_i - \tilde{x})^\top D_{\tilde{x}}^2 F(\tilde{y}|\tilde{x})(\tilde{X}_i - \tilde{x})b_i(\tilde{x})K_h(\tilde{X}_i - \tilde{x})]$$
$$= \mathrm{tr}(D_{\tilde{x}}^2 F(\tilde{y}|\tilde{x})\mathbb{E}[(\tilde{X}_i - \tilde{x})(\tilde{X}_i - \tilde{x})^\top b_i(\tilde{x})K_h(\tilde{X}_i - \tilde{x})])$$
$$= h^2\mu_2 g(\tilde{x})\mathrm{tr}(D_{\tilde{x}}^2 F(\tilde{y}|\tilde{x})) + o_p(h^2),$$

we have

$$J_2 = \frac{1}{2}h^2\mu_2\mathrm{tr}(D_{\tilde{x}}^2 F(\tilde{y}|\tilde{x}))g(\tilde{x}) + o_p(h^2). \tag{A.15}$$

Finally, by applying the expansion argument routinely, we get

$$J_3 = g(\tilde{x}) + o_p(1). \tag{A.16}$$

$\square$

*Proof of Proposition 4.6 (Cai, 2002; Salha, 2006).* Combining Lemmas B.3 and B.4 with equation A.8, Assumption 4.5 gives the result. $\square$

*Proof of Corollary 4.7 (Cai, 2002).* Given $\tilde{x}$, Proposition 4.6 implies uniform convergence of $\widehat{F}(\cdot|\tilde{x})$ to $F(\cdot|\tilde{x})$ in probability (Tucker, 1967, p.127-128) since $F(\cdot|\tilde{x})$ is a CDF. That is,

$$\sup_{\tilde{y} \in \mathbb{R}} \left|\widehat{F}(\tilde{y}|\tilde{x}) - F(\tilde{y}|\tilde{x})\right| \to 0 \text{ in probability.}$$

Given $\varepsilon > 0$, let $\delta = \delta(\varepsilon) := \min\{\beta - F(Q_\beta(\tilde{x}) - \varepsilon|\tilde{x}), F(Q_\beta(\tilde{x}) + \varepsilon|\tilde{x}) - \beta\}$. Note that $\delta > 0$ due to the uniqueness of the quantile. We have

$$\mathbb{P}\left(\left|\widehat{Q}_\beta(\tilde{x}) - Q_\beta(\tilde{x})\right| > \varepsilon\right) \leq \mathbb{P}\left(\left|F(\widehat{Q}_\beta(\tilde{x})|\tilde{x}) - \beta\right| > \delta\right)$$

$$\leq \mathbb{P}\left(\sup_{\tilde{y} \in \mathbb{R}} \left|\widehat{F}(\tilde{y}|\tilde{x}) - F(\tilde{y}|\tilde{x})\right| > \delta\right).$$

The uniform convergence of $\widehat{F}(\cdot|\tilde{x})$ in probability gives the result. $\square$

*Proof of Corollary 4.8.* From the definition of $\hat{C}^{\alpha}_{t-1}$ in equation 3, we have

$$\mathbb{P}\left(Y_t \in \hat{C}^{\alpha}_{t-1}(X_t)\Big| X_t\right) = \mathbb{P}\left(\tilde{Y}_{n+1} \in \left[\widehat{Q}_{\beta^*}(\tilde{X}_{n+1}), \widehat{Q}_{1-\alpha+\beta^*}(\tilde{X}_{n+1})\right] \Big| \tilde{X}_{n+1}\right)$$
$$= F\left(\widehat{Q}_{1-\alpha+\beta^*}(\tilde{X}_{n+1})\Big|\tilde{X}_{n+1}\right) - F\left(\widehat{Q}_{\beta^*}(\tilde{X}_{n+1})\Big|\tilde{X}_{n+1}\right).$$

By Theorem 4.7, we have the consistency of $\widehat{Q}_{\beta}$ for all $\beta \in (0,1)$. On that, the continuous mapping theorem and Assumption 4.3 gives

$$F\left(\widehat{Q}_{1-\alpha+\beta^*}(\tilde{X}_{n+1})\Big|\tilde{X}_{n+1}\right) - F\left(\widehat{Q}_{\beta^*}(\tilde{X}_{n+1})\Big|\tilde{X}_{n+1}\right)$$
$$\to F\left(Q_{1-\alpha+\beta^*}(\tilde{X}_{n+1})\Big|\tilde{X}_{n+1}\right) - F\left(Q_{\beta^*}(\tilde{X}_{n+1})\Big|\tilde{X}_{n+1}\right)$$
$$= 1 - \alpha,$$

where the convergence is in probability. □

*Proof of Theorem 4.9.* From the definition of $\widehat{C}^{\alpha}_{t-1}$ in equation 3, we have

$$\left|\mathbb{P}\left(Y_t \in \widehat{C}^{\alpha}_{t-1}(x)\Big| X_t = x\right) - (1-\alpha)\right|$$
$$= \left|\mathbb{P}\left(\tilde{Y}_{n+1} \in \left[\widehat{Q}_{\beta^*}(\tilde{x}), \widehat{Q}_{1-\alpha+\beta^*}(\tilde{x})\right]\Big| \tilde{X}_{n+1} = \tilde{x}\right) - (1-\alpha)\right|$$
$$= \left|F\left(\widehat{Q}_{1-\alpha+\beta^*}(\tilde{x})\Big|\tilde{X}_{n+1} = \tilde{x}\right) - F\left(\widehat{Q}_{\beta^*}(\tilde{x})\Big|\tilde{X}_{n+1} = \tilde{x}\right) - (1-\alpha)\right|$$
$$\leq \left|\widehat{F}\left(\widehat{Q}_{1-\alpha+\beta^*}(\tilde{x})\Big|\tilde{X}_{n+1} = \tilde{x}\right) - (1-\alpha+\beta^*)\right| + \left|\widehat{F}\left(\widehat{Q}_{\beta^*}(\tilde{x})\Big|\tilde{X}_{n+1} = \tilde{x}\right) - \beta^*\right|$$
$$+ \left|F\left(\widehat{Q}_{1-\alpha+\beta^*}(\tilde{x})\Big|\tilde{X}_{n+1} = \tilde{x}\right) - \widehat{F}\left(\widehat{Q}_{1-\alpha+\beta^*}(\tilde{x})\Big|\tilde{X}_{n+1} = \tilde{x}\right)\right|$$
$$+ \left|F\left(\widehat{Q}_{\beta^*}(\tilde{x})\Big|\tilde{X}_{n+1} = \tilde{x}\right) - \widehat{F}\left(\widehat{Q}_{\beta^*}(\tilde{x})\Big|\tilde{X}_{n+1} = \tilde{x}\right)\right|$$
$$\leq 2\Delta(\tilde{x}) + \mathcal{O}_p((nh^w)^{-1/2} + h^2),$$

where the last inequality comes from equation A.8 and the definition of the discrete gap $\Delta$. □

## C  ADDITIONAL REAL DATA EXPERIMENTS

### C.1  WIND/SOLAR DATA EXPERIMENT RESULTS

We provide a more detailed description of the results for the solar and wind datasets introduced in Section 5. Figures A.1 and A.2 illustrate the rolling coverage and interval width results for the solar and wind datasets, respectively. As described in Section 5, KOWCPI consistently achieves the narrowest intervals while maintaining valid coverage. For qualitative explanations, we also include Figure A.3 that demonstrates the performance of KOWCPI, SPCI, and ACI on the Solar dataset.

### C.2  AAPL DAILY STOCK PRICE

We compare KOWCPI with baseline methods using Apple's daily closing stock price data from January 1, 2020, to December 12, 2022. This publicly available dataset can be accessed on Kaggle (https://www.kaggle.com/datasets/paultimothymooney/stock-market-data). The goal is to construct confidence intervals for the daily closing prices. The first 80% of the data is used for training, with the remaining 20% reserved for evaluation. We can observe from Table A.1 that KOWCPI attains the narrowest interval.

## D  SYNTHETIC DATA ANALYSIS

### D.1  HETEROSKEDASTIC MIXTURE MODEL

To evaluate the robustness of KOWCPI under heteroskedastic conditions, we conduct simulations using a heteroskedastic mixture model. This model incorporates an AR(1) component, a GARCH(1,1)

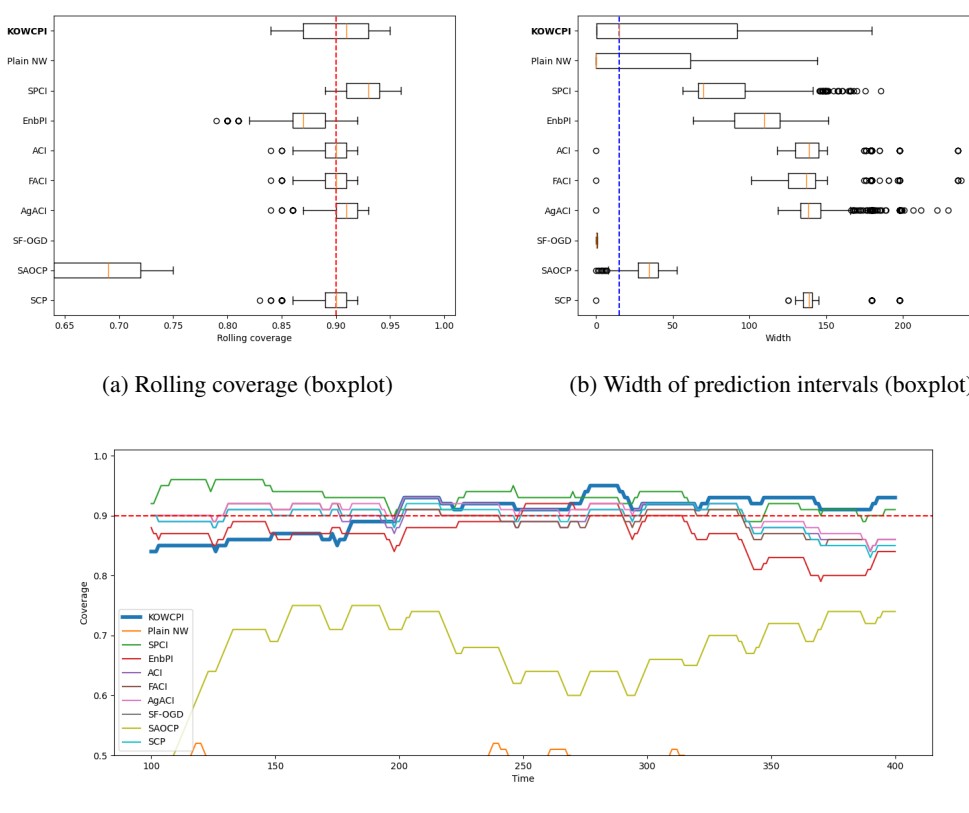

(a) Rolling coverage (boxplot)

(b) Width of prediction intervals (boxplot)

(c) Rolling coverage over prediction time indices

Figure A.1: Rolling coverage and width comparison on the solar dataset by different methods.

Table A.1: Empirical marginal coverage and interval widths for confidence intervals of AAPL closing prices, with a target coverage of 90%. Standard deviations are calculated under three independent trials.

|        | Coverage | Width |
|--------|----------|-------|
| KOWCPI | 0.912 (2.3e-3) | 15.74 (1.2e-1) |
| SPCI   | 0.952 (3.3e-3) | 19.39 (2.3e-1) |
| EnbPI  | 0.912 (8.7e-3) | 38.67 (1.5e-1) |
| ACI    | 0.871 (1.7e-3) | 44.84 (8.7e-2) |
| FACI   | 0.891 (5.2e-3) | 43.93 (1.7e-1) |
| AgACI  | 0.878 (1.1e-2) | 43.19 (2.6e-1) |
| SAOCP  | 0.619 (7.1e-4) | 17.90 (4.3e-2) |
| SCP    | 0.796 (2.0e-3) | 36.60 (1.0e-1) |

structure for time-varying variance, and an additional small Gaussian noise term. The model is defined as

$$Y_t = 0.8Y_{t-1} + \sigma_t \epsilon_t + \xi_t,$$
$$\sigma_t^2 = 0.1 + 0.3Y_{t-1}^2 + 0.6\sigma_{t-1}^2,$$
$$\epsilon_t \overset{iid}{\sim} N(0,1), \quad \xi_t \overset{iid}{\sim} N(0, 0.1^2).$$

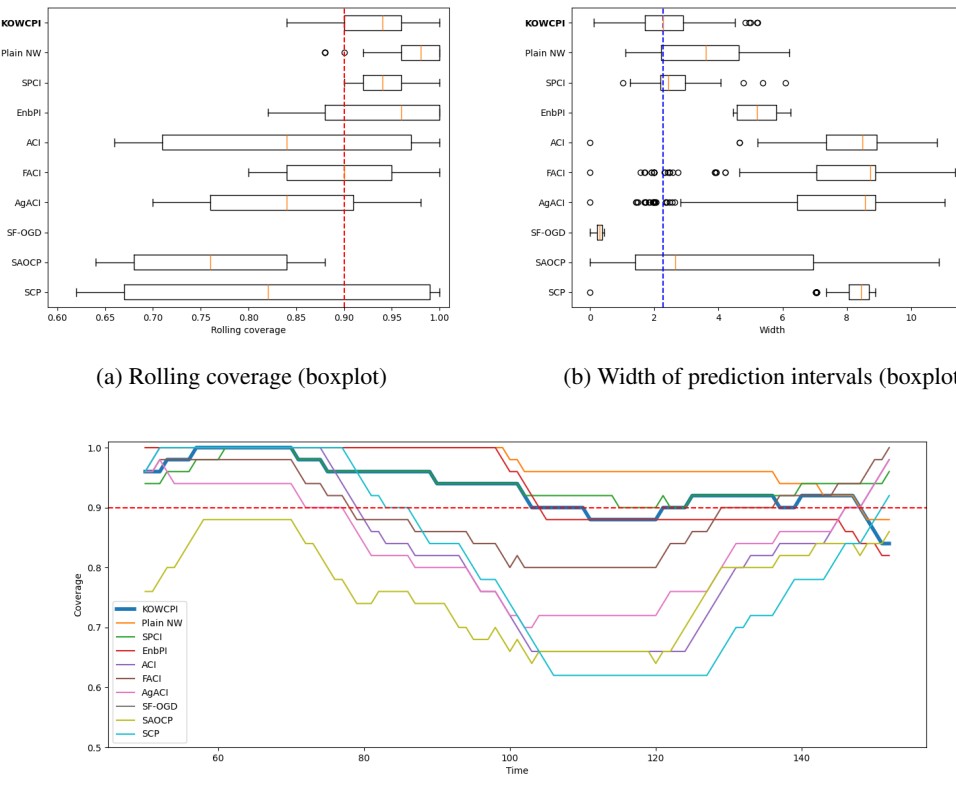

(a) Rolling coverage (boxplot)   (b) Width of prediction intervals (boxplot)

(c) Rolling coverage over prediction time indices

Figure A.2: Rolling coverage and width comparison on the wind dataset by different methods.

This mixture model generates irregular, large volatility bursts, as evidenced in the simulated sample paths. Such extreme variations make conformal prediction challenging, as they require rapid adaptation to maintain valid coverage while avoiding overly wide intervals.

We simulate five independent paths of the model and evaluate the performance of KOWCPI against baseline methods, with a target coverage of 90%. Unlike methods such as SPCI and SAOCP, which often overreact to volatility changes by producing excessively wide intervals and struggle to recover quickly after a burst, KOWCPI effectively adapts to these changes using its adaptive weighting mechanism. The results for each sample path are summarized in Table A.2.

### D.2 NONSTATIONARY TIME SERIES

We also consider a model with strong seasonality, clearly representing non-stationarity:

$$Y_t = \log(t') \sin\left(\frac{2\pi t'}{12}\right) \left(|\beta^\top X_t| + |\beta^\top X_t|^2 + |\beta^\top X_t|^3\right)^{1/4} + \epsilon_t,$$

where $t' = \mathrm{mod}(t, 12)$ introduces a seasonal component with a 12-period cycle, and $X_t = [Y_{t-100}, \ldots, Y_{t-1}]^\top$ represents features of lagged values. The noise term $\epsilon_t$ follows an AR(1) process, given by $\epsilon_t = 0.6\epsilon_{t-1} + \xi_t$, with $\xi_t \sim N(0,1)$. Table A.3 demonstrates that KOWCPI performs best in general among methods that achieve valid coverage of 90%. As KOWCPI actively leverages reweighting to assign higher weights to recent residuals, it demonstrates the ability to quickly adapt to changes in volatility.

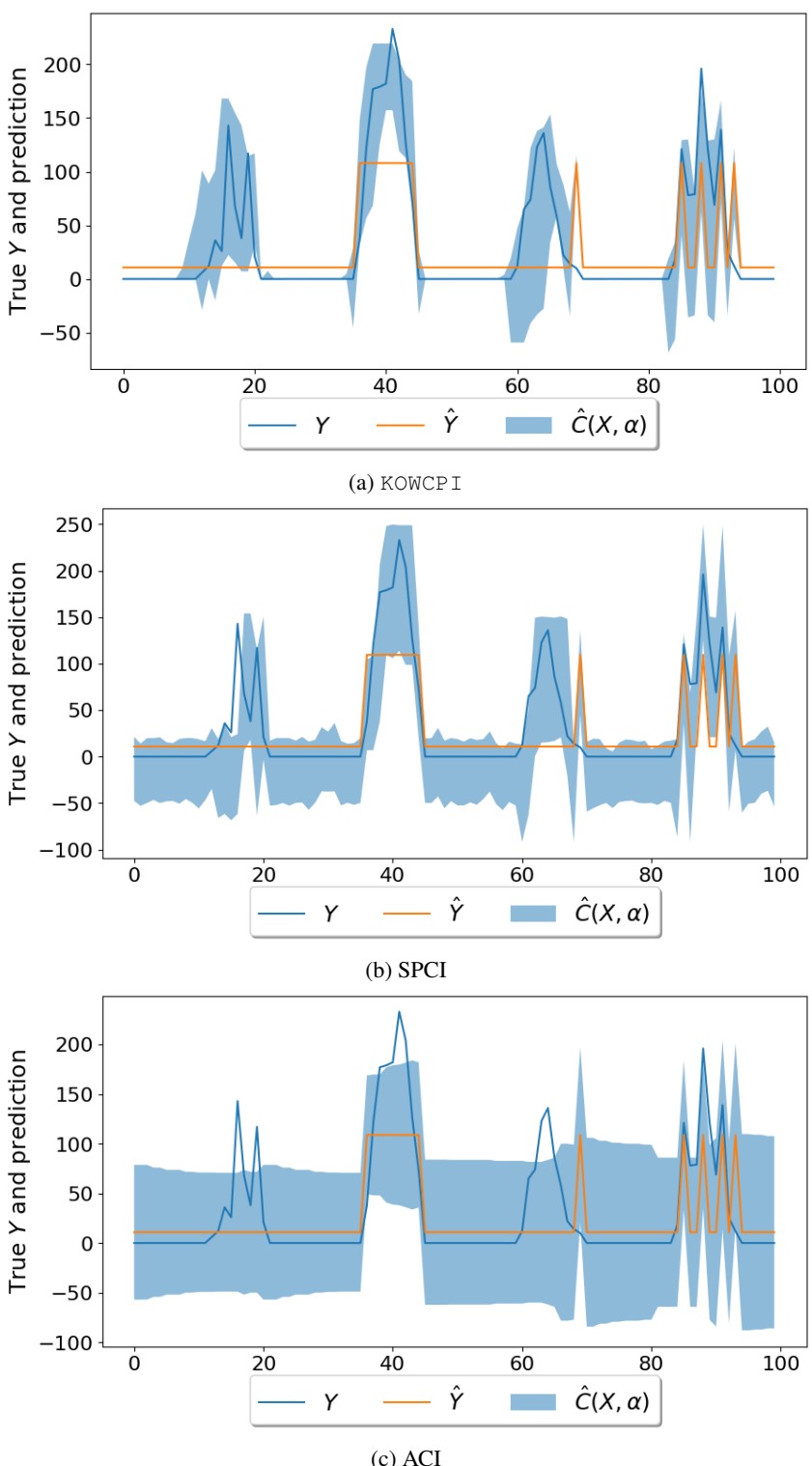

Figure A.3: Comparison of prediction intervals generated by KOWCPI, SPCI, and ACI on the Solar dataset.

Table A.2: Empirical marginal coverage and interval widths from simulations using a heteroskedastic mixture model, with a target coverage of 90%. Here, C and W denote the empirical marginal coverage and average interval width, respectively.

|  | Path 1 | | Path 2 | | Path 3 | | Path 4 | | Path 5 | |
|---|---|---|---|---|---|---|---|---|---|---|
|  | C | W | C | W | C | W | C | W | C | W |
| KOWCPI | 0.91 | 4.61 | 0.89 | 5.57 | 0.90 | 11.44 | 0.93 | 8.84e3 | 0.92 | 23.09 |
| SPCI | 0.99 | 8.68 | 0.89 | 5.85 | 0.97 | 18.46 | 0.90 | 8.53e3 | 1.00 | 99.22 |
| EnbPI | 0.93 | 5.47 | 0.87 | 5.56 | 0.91 | 15.60 | 0.91 | 9.14e3 | 0.96 | 74.62 |
| ACI | 0.93 | 5.21 | 0.92 | 6.99 | 0.92 | 14.11 | 0.89 | 1.79e4 | 0.95 | 27.84 |
| FACI | 0.92 | 5.11 | 0.92 | 7.10 | 0.92 | 13.75 | 0.89 | 1.88e4 | 0.92 | 24.28 |
| AgACI | 0.93 | 5.20 | 0.91 | 6.83 | 0.93 | 13.17 | 0.88 | 1.58e4 | 0.92 | 25.83 |
| SAOCP | 0.79 | 3.58 | 0.82 | 4.62 | 0.72 | 7.39 | 0 | 36.1 | 0.67 | 10.87 |
| SCP | 0.93 | 5.17 | 0.91 | 6.73 | 0.93 | 14.19 | 0.84 | 1.06e4 | 0.97 | 29.23 |

Table A.3: Empirical marginal coverage and interval widths from nonstationary time-series simulations, with a target coverage of 90%. Standard deviations are calculated under five independent trials.

|  | Coverage | Width |
|---|---|---|
| KOWCPI | 0.90 (1.2e-3) | 11.41 (2.3e-2) |
| SPCI | 0.91 (2.7e-3) | 11.73 (3.1e-2) |
| EnbPI | 0.86 (1.2e-2) | 10.45 (1.8e-2) |
| ACI | 0.90 (1.1e-3) | 12.57 (8.7e-3) |
| FACI | 0.90 (4.1e-3) | 12.65 (1.2e-2) |
| AgACI | 0.90 (2.2e-3) | 12.71 (1.4e-2) |
| SAOCP | 0.82 (9.4e-4) | 8.89 (3.2e-3) |
| SCP | 0.90 (2.8e-3) | 12.50 (4.1e-2) |

## E  ADAPTIVE WINDOW LENGTH SELECTION

In this section, we explore an adaptive selection of $w$, where $w$ is no longer treated as a fixed hyperparameter but is instead dynamically adjusted for each time step. In KOWCPI, the window length $w$ originally serves as a hyperparameter that requires tuning. To alleviate the burden of manual tuning and introduce a more data-driven approach, we implement an adaptive selection process for $w$ based on a two-sample test on the residual distributions.

At each time step $t$, we compare the distributions two blocks of residuals using the two-sample Kolmogorov-Smirnov test: One block contains the most recent $w$ residuals $(\hat{\varepsilon}_{t-1}, \ldots, \hat{\varepsilon}_{t-w})$, and another block consists of the $w$ residuals immediately preceding, $(\hat{\varepsilon}_{t-w-1}, \ldots, \hat{\varepsilon}_{t-2w})$. We then select the smallest $w$ for which the p-value drops below, e.g., 0.01.

While this is a simple preliminary approach, it allows for a data-driven and adaptive selection of $w$ without requiring additional hyperparameter tuning. Through experiments on the real data, we have confirmed that this method achieves comparable performance to $w$ values pre-selected by cross-validation (See Table A.4). Figure A.4 illustrates how the chosen window size changes over time on the Wind dataset.

Table A.4: Comparison of `KOWCPI` on real datasets using pre-fixed window lengths selected by cross-validation versus adaptive window selection based on the two-sample KS test. Target coverage is 90%, and standard deviation is derived across five independent trials.

|  | Electric | | Wind | | Solar | |
|---|---|---|---|---|---|---|
|  | Coverage | Width | Coverage | Width | Coverage | Width |
| Fixed $w$ | 0.90 (2.3e-3) | 0.23 (1.5e-3) | 0.91 (2.8e-3) | 2.41 (3.2e-2) | 0.90 (1.2e-3) | 48.8 (9.4e-1) |
| Adaptive $w$ | 0.92 (3.0e-3) | 0.22 (1.3e-3) | 0.90 (4.4e-3) | 2.44 (2.7e-2) | 0.90 (1.3e-3) | 50.6 (1.1e+0) |

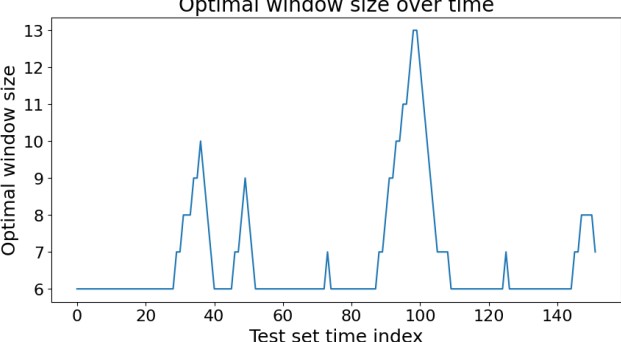

Figure A.4: Dynamic adjustment of the window size ($w$) for each prediction step on the Wind dataset, using the adaptive selection process based on the two-sample KS test.

