# OpenReview forum: "Kernel-based Optimally Weighted Conformal Time-Series Prediction"
_ICLR.cc/2025/Conference — ICLR 2025 Poster_

### Official Review · Reviewer_WMCc · 2024-10-20

**Soundness:** 3
**Presentation:** 3
**Contribution:** 2
**Rating:** 6
**Confidence:** 3

**Summary:**

The paper presents a method to construct prediction intervals for time-series data, its main contribution being using RNW estimators for quantile regression. The paper builds upon a large body of work on the topic and offers a new algorithm under mild assumptions.

**Strengths:**

- The paper is very well-written and easy to follow.
- The algorithm that leverages the RNW estimator is innovative.
- Theory section (as far as I have understood) is correct and well-argued.
- The experiment comparison is comprehensive (includes almost all the baselines in the space. thanks for doing all this work).

**Weaknesses:**

**1. Differentiation from SPCI.**

One concern I have is that this approach feels very similar to SPCI both in terms of algorithm (using an estimator to model residuals) and guarantee / proof technique (relies on the convergence property of the estimator). In terms of experiment results, KOWCPI also does not significantly outperform SPCI (in figure 2(c), there is a significant chunk of time where SPCI is more valid).

Can you add some more explanation on how your method is different from SPCI, in terms of theoretical guarantees, assumptions on data, and performance? It will not only improve clarity, but also help readers in practice know which method to choose for their specific applications.

**2. Experiment results & analysis**

I would strongly encourage the authors to add more details to their experiment results, including:

- Add standard deviations to table 1.
- Explain the results in table 1. What are the rationale of highlighting the first row? You should also highlight other methods that achieve the same coverage, and explain  why lower width was not highlighted.
- Add experiments for one or more values of $\alpha$ to demonstrate consistent performance.
- Provide some qualitative examples to show why/how the KOWCPI confidence bands are better.
- Elaborate on figure 2(d), perhaps adding a further study/analysis on weights. The remark in line 494-496 is not very convincing when only one example is provided.

If these concerns are addressed, I'm happy to increase my score.

**Questions:**

See weaknesses.

---

> ### Author Response · Authors · 2024-11-24
>
> Dear Reviewer WMCc,
>
> Thank you for your insightful feedback and thoughtful suggestions, which have greatly helped us refine our work. In the following, we address your comments and provide clarifications to the points you raised.
>
> **Weakness 1: Differentiation from SPCI**
>
> Thank you for bringing up this important point. Both of the methods utilize quantile regression, rather than the empirical quantiles, to predict future quantile of residuals, and thus the consistency of the quantile estimator plays a critical role in the coverage guarantee. The main difference lies in KOWCPI’s incorporation of adaptive reweighting of historical residuals through a nonparametric kernel-based approach. Due to the smoothing effect of the kernel, this allows KOWCPI to construct more stable confidence intervals, particularly under challenging scenarios such as distribution shifts or sudden volatility spikes. In contrast, SPCI relies on quantile regression forests (QRF), which are known to be noisy when the data is imbalanced, often resulting in spiked intervals during such conditions, as observed in our experiments (e.g., Path 5 in Appendix D.1. Please refer to the plot in the [Link](https://anonymous.4open.science/r/KOWCPI_figures-97BD/path_5_SPCI.png)).
>
> **Theory and Assumptions**
> -  SPCI assumes that the decay of the estimated weights from the QRF estimator is linear, i.e., $O(1/n)$, for its asymptotic conditional guarantee, though it is only known to be $o(1)$ in the literature [21]. This assumption is equivalent to the discrete gap $\Delta$ introduced in Equation (17) decaying linearly, a requirement not imposed in KOWCPI. Additionally, the consistency of QRF requires the covariates (in this case, $\tilde X$) to be uniformly distributed over the compact support, which is unlikely for residuals, which are expected to center around zero with a "well-performing" point predictor.
> - KOWCPI, on the other hand, does not impose assumptions on the estimated weights. The RNW estimator used in KOWCPI inherently guarantees consistency under standard conditions without further assumptions about the estimated weights themselves. While both QRF and RNW estimators assume smoothness of the conditional CDF for their consistency, the strong mixing assumption for the residual process in KOWCPI is arguably more realistic than the uniformity assumption required by QRF (Please refer to Point 1 in the common response). Furthermore, unlike QRF, the RNW estimator has a derivable convergence rate, which enabled the development of theoretical results such as Theorem 4.9 in our paper.
>
> **Performance Comparison**
> - *Real data experiments.* While the original paper may not have fully illustrated KOWCPI’s advantages, the results consistently show that KOWCPI achieves narrower interval widths (considering the standard deviations of widths) across all real data experiments. KOWCPI avoids the over-coverage issue often observed in SPCI, which results in unnecessarily wide intervals.  In the revision, we have included an additional real data experiment using stock price data, where KOWCPI once again outperformed SPCI.
> - *Handling heteroskedasticity.* For datasets with heteroskedasticity, SPCI requires an additional step to estimate the standard deviation of residuals to maintain competitive performance. In contrast, KOWCPI inherently accounts for such variations through its adaptive reweighting mechanism, eliminating this extra complexity.
> - *Adaption to volatility.* When a burst of volatility occurs, SPCI produces wider intervals and takes longer to recover narrow intervals even after the sample path stabilizes, whereas KOWCPI adapts quickly to these changes.  To illustrate this, we conducted additional simulations using an AR(1) + GARCH(1,1) mixture model. The results, presented in Appendix D.2 and Table 2 of the common response, demonstrate that KOWCPI consistently outperforms SPCI by achieving narrower intervals with valid coverage while adapting more effectively to volatility. For further details including this scenario, and qualitative explanations pertaining to the reweighting mechanism, please see our following response to Weaknesses 2d & 2e.
>
> [21] Meinshausen, N., Quantile regression forests. *Journal of Machine Learning Research*, 7:983–999, 2006.

---

> ### Author Response · Authors · 2024-11-24
>
> **Weakness 2a: standard deviations**
>
> Thank you for your detailed suggestions. In the revision, we have included standard deviations for each metric in all result tables, including those for the additional experiments.
>
> **Weakness 2b: Table 1 highlighting**
>
> We apologize for the confusion that the table mistakenly used bold text to highlight only our method (KOWCPI) rather than the best-performing methods. In the revision, we have removed highlights from the table to avoid any potential misinterpretation.
>
> **Weakness 2c: different significance levels**
>
> According to your suggestion, we have conducted additional experiments on the Wind dataset for 80% and 95% confidence levels. The results confirm that KOWCPI maintains valid coverage and achieves the narrowest intervals, consistent with the results for 90% coverage presented in the paper. We commit to include a comprehensive list of various target coverages with other datasets in the final revision.
>
> It is noteworthy that the same hyperparameters from the 90% target experiment were applied without any additional tuning; that is, KOWCPI does not require additional parameter tuning to succeed across diverse target coverage levels.
>
> Table 3: Comparison of our KOWCPI and other baseline methods on the Wind dataset with target coverages of 80% and 95%. Standard deviations are calculated across five independent trials.
>
> | Method  | Coverage (95% target) | Width (95% target) | Coverage (80% target) | Width (80% target) |
> |---------|------------------------------|--------------------------|-----------------------------|--------------------------|
> | KOWCPI  | 0.95 (4.1e-3)               | 2.74 (3.1e-2)           | 0.81 (5.0e-3)              | 1.75 (8.1e-3)           |
> | SPCI    | 0.97 (2.9e-3)               | 2.89 (1.8e-2)           | 0.84 (3.7e-3)              | 1.87 (4.6e-3)           |
> | EnbPI   | 0.93 (7.7e-3)               | 5.64 (2.8e-2)           | 0.87 (1.2e-2)              | 4.77 (1.7e-2)           |
> | ACI     | 0.93 (1.1e-2)               | 9.05 (3.1e-2)           | 0.76 (2.2e-2)              | 6.68 (1.8e-2)           |
> | FACI    | 0.95 (6.5e-3)               | 8.75 (3.3e-2)           | 0.82 (3.9e-3)              | 4.78 (4.3e-2)           |
> | AgACI   | 0.95 (2.1e-3)               | 8.63 (1.8e-2)           | 0.82 (6.1e-3)              | 4.72 (1.5e-2)           |
> | SAOCP   | 0.80 (1.2e-3)               | 4.43 (6.2e-3)           | 0.71 (9.4e-4)              | 3.72 (1.0e-2)           |
> | SCP     | 0.93 (3.1e-3)               | 8.68 (1.3e-2)           | 0.78 (3.4e-3)              | 6.99 (2.1e-2)           |

---

> ### Author Response · Authors · 2024-11-24
>
> **Weaknesses 2d: qualitative explanations & 2e: discussion on weights**
>
> Thank you for raising these insightful points. To give rationales for KOWCPI’s strong performance, we believe a discussion on the behavior of the weights in the RNW estimator is essential. Therefore, we address both comments together.
>
> **Behavior of weights in RNW estimator**
>
> The RNW estimator incorporates reweighting with adjustment weights to reduce bias. Figure 2(d) aimed to illustrate that the final weights $\hat{W}_i$ align with intuition, assigning significantly larger weights to residual blocks from more recent past data, particularly the most recent block. This property is central to interpreting KOWCPI’s results, but we recognize that our original discussion was overly simplistic. We are grateful for the opportunity to expand on this point.
>
> You can find weight plots for the Solar and Wind datasets ([link](https://anonymous.4open.science/r/KOWCPI_figures-97BD/log_weights.png)), which consistently show that recent residuals receive substantially higher weights. In the final revision, we will incorporate a more detailed and quantitative discussion on the weights to strengthen the following arguments.
>
> **Why KOWCPI outperforms?**
>
> The reweighting properties of KOWCPI help explain its superior performance, particularly in cases like the Solar dataset, where the volatility periodically alternates between high and low levels. KOWCPI adapts swiftly to these changes, generating very narrow prediction intervals in low-volatility regions while dynamically following the trajectory of $Y$ during high-volatility regions. In contrast, methods like SPCI and ACI tend to produce intervals of relatively uniform width and do not stray far from their point estimators. Unlike empirical quantile-based methods or QRF-based SPCI, KOWCPI leverages reweighting to assign higher weights to recent residuals, enabling it to respond quickly to changes in volatility.
>
> (Please refer to the links for [KOWCPI](https://anonymous.4open.science/r/KOWCPI_figures-97BD/Solar_KOWCPI.png), [SPCI](https://anonymous.4open.science/r/KOWCPI_figures-97BD/Solar_SPCI.png), and [ACI](https://anonymous.4open.science/r/KOWCPI_figures-97BD/Solar_ACI.png) behavior plots on the Solar dataset, where KOWCPI demonstrates its adaptability compared to the static intervals of other methods.)
>
> This adaptability is also evident in the AR+GARCH mixture model experiment (Appendix D.1). As mentioned in our response to Weakness 1, after volatility bursts occur, KOWCPI recovers from these bursts significantly faster than other methods. For instance, in Table A.2, Path 3 and Path 5 exemplify this behavior. Path 3 ([link](https://anonymous.4open.science/r/KOWCPI_figures-97BD/path_3.png)), which experiences high variance in the early training phase and early test phase, shows that SPCI gradually narrows its intervals, while KOWCPI adjusts its interval widths almost immediately to match the real-time volatility (Plots: [KOWCPI](https://anonymous.4open.science/r/KOWCPI_figures-97BD/path_3_KOWCPI.png), [SPCI](https://anonymous.4open.science/r/KOWCPI_figures-97BD/path_3_SPCI.png), [ACI](https://anonymous.4open.science/r/KOWCPI_figures-97BD/path_3_ACI.png)). Similarly, for Path 5 ([link](https://anonymous.4open.science/r/KOWCPI_figures-97BD/path_5.png)), where most methods suffer from severe overcoverage following a burst that happens right before the test phase, SPCI fails to recover and continues to produce overly conservative intervals. In contrast, KOWCPI quickly narrows its intervals, achieving an average narrower width (Plots: [KOWCPI](https://anonymous.4open.science/r/KOWCPI_figures-97BD/path_5_KOWCPI.png), [SPCI](https://anonymous.4open.science/r/KOWCPI_figures-97BD/path_5_SPCI.png), [ACI](https://anonymous.4open.science/r/KOWCPI_figures-97BD/path_5_ACI.png)). However, this case also highlights a limitation of KOWCPI: during the earliest test steps, KOWCPI briefly fails to maintain coverage due to its aggressive adaptation to rapidly changing conditions. We have organized this discussion and added it to the Results paragraph of Experiments section in the revision (Lines 474-485).
>
> **Theoretical trade-off**
>
> Assigning higher weights to recent residuals enables KOWCPI to adapt quickly, producing narrower intervals and avoiding over-coverage. From a theoretical perspective, however, our bounds for marginal and conditional coverage are involved with the discrete gap $\Delta$, which is the uniform bound of the weights (Proposition 4.1, Theorem 4.9). Therefore, this approach introduces a trade-off: such reweighting mechanism that allows rapid adaptation can create coverage gaps, especially in extreme cases where the data distribution changes abruptly. Nonetheless, despite the precise decay rate of $\Delta$ being an open question, the consistency of the RNW estimator still implies the validity of KOWCPI’s conditional coverage guarantees.

---

> > ### Comment · Reviewer_WMCc · 2024-11-25
> > **Thank you for the thoughtful discussion**
> >
> > Thanks to the authors for the additional details and edits.
> >
> > My concerns are addressed and I have increased my score.

---

> > > ### Author Response · Authors · 2024-11-25
> > >
> > > We greatly appreciate your recognition and are glad that our responses and revisions were able to address your concerns. Thank you so much for your support!

---

### Official Review · Reviewer_75Ps · 2024-10-24

**Soundness:** 2
**Presentation:** 3
**Contribution:** 2
**Rating:** 6
**Confidence:** 3

**Summary:**

This paper proposes a conformal prediction method for time-series data using the Reweighted Nadaraya-Watson estimator. The authors provide non-asymptotic marginal coverage gap in Proposition 4.1 and conditional coverage guarantees under strong mixing conditions in Theorem 4.9. Its efficiency is demonstrated through experiments on three real-world datasets.

**Strengths:**

1. The paper is well-written and easy-to-follow.
2. The considered problem is important and well-motivated.
3. The proposed KOWCPI is efficient in various real datasets.

**Weaknesses:**

1. Regarding the experiments, the author only provides real datasets to demonstrate the efficiency of marginal and conditional coverage without offering information about dependency. However, I believe it would be beneficial to include additional synthetic experiments to explore different dependency structures and more comprehensively assess the scope and validity of the methods.

2. While a theoretical guarantee is provided, the bound heavily depends on the estimation of weights and the strong mixing assumptions.

**Questions:**

1. Regarding the theoretical results, one of my concerns is could the author provide theoretical guarantees for other types of dependent time-series beyond strong mixing assumption or explain the difficulties under other dependent conditions?

2. Related works and experiments:
- Could the author provide more details about the benchmarks and related works to help readers unfamiliar with online conformal prediction understand why KOWCPI outperforms other methods? For example, providing a brief summary of the key differences between KOWCPI and each baseline method, highlighting why these differences might lead to improved performance may be helpful.  You can also include a table to compare the main features of each method if possible.
- Besides, the comparison methods in experiments seem to be incomplete. For example, the conformal PID in [1], the BCI in [2] and the decay-OGD in [3] should be considered.

4. Optimal choice of window length $\omega$:
 - Is it possible to design some strategy to determine optimal $\omega$ values that adapt to real-time changes in data volatility or distribution patterns, thereby further enhancing the KOWCPI? For example, using change point detection algorithms or online learning techniques to dynamically adjust the window length.

References:

[1] Angelopoulos A, Candes E, Tibshirani R J. Conformal PID control for time series prediction[J]. NeurIPS, 2023, 36.

[2] Yang Z, Candès E, Lei L. Bellman Conformal Inference: Calibrating Prediction Intervals For Time Series[J]. arXiv, 2024.

[3] Angelopoulos A, Barber R F, Bates S. Online conformal prediction with decaying step sizes. ICML, 2024

---

> ### Author Response · Authors · 2024-11-23
>
> Dear Reviewer 75Ps,
>
> We deeply value your feedback and appreciate the opportunity to address your concerns in this response.
>
> **Weakness 1: synthetic experiments with dependency structures**
>
> Thank you for your constructive suggestions to strengthen KOWCPI’s broad applicability. We have constructed additional experiments with (i) synthetic dataset of mixture model of AR(1) + GARCH(1,1) with heteroskedascity, given as
> $$
> 	Y_t = 0.8Y_{t-1} + \sigma_t \epsilon_t + \xi_t, \quad \sigma_t^2 = 0.1 + 0.3Y_{t-1}^2 + 0.6 \sigma_{t-1}^2, \quad \epsilon_t \stackrel{iid} \sim N(0,1), \quad \xi_t \stackrel{iid} \sim N(0, 0.1^2),
> $$
>
> and (ii) nonstationary time series with seasonal trend, given as
> $$
> 	Y_t = \log(t’) \sin \left(\frac{2 \pi t’}{12} \right) (|\beta^T X_t| + |\beta^T X_t|^2 + |\beta^T X_t|^3)^{1/4} + \epsilon_t, \quad \epsilon_t \sim AR(1).
> $$
>
> We have newly included in Appendix D of the revision for additional experiments with these synthetic datasets. Both with strong dependency structure with clear trend, and under heteroskedacity with irregular volatility bursts, KOWCPI remains to obtain valid coverage and the narrowest width compared to baseline methods. Please refer to Appendix D of the revision for detailed results.
>
> **Weakness 2 and Question 1: theoretical guarantee with strong mixing assumption and weight estimation**
>
> We acknowledge the limitation in our coverage gap bounds, as you mentioned. However, we would like to emphasize that the strong mixing assumption on residuals (not on the data itself) is not as restrictive as it may appear, as highlighted in Note 1 of the common response and also in the paper (Lines 320-323). This assumption, standard in time series analysis, ensures that the dependence in residuals decays at a sufficient rate and can often hold even when the original time series data exhibit non-stationary behaviors.
>
> Regarding the estimation of weights, we would like to share the following uniform bound result for the adjustment weights (excluding boundary data), derived based on [9]:
> $$
> 	\sup_{x \in \mathbb{S}} \Big| p_i(x) - \frac{1}{n} \Big| = O\left(h + \sqrt{\frac{\log n}{nh^w}}\right) \quad \text{almost surely},
> $$
> for any compact subset $\mathbb{S}$ where the density is uniformly bounded from below on $\mathbb{S}$. This result demonstrates that the adjustment weights uniformly behave around $1/n$, providing theoretical support for the stability of KOWCPI, while it may also suggest the rate at which the discrete gap decays. However, we note that a precise bound of the final weight $\hat W_i$, which is reweighted by kernel values, remains an open question.
>
> Considering the other types of dependency, if we replace Assumption 4.2 for mixing of the residual process with the following:
>
> **Assumption 4.2’.** *$(V\_i=(\tilde X\_i, \tilde Y\_i))\_{i=1}^\infty$ is absolutely regular, and there exists $\delta \in [0,1)$ such that $\sum_{j} j^2 \beta(j)^{\delta/(1 + \delta)} < \infty$,*
>
> the RNW estimator achieves consistency with the same rate as under the strong-mixing assumption, namely $O(h^2 + (nh^w)^{-1/2})$ ([10]). Therefore, this also leads to valid asymptotic coverage for KOWCPI. However, we note that this assumption on absolute regularity is generally stronger than Assumption 4.2. For this reason, we chose to focus on the results derived under the weaker strong-mixing framework in the paper.
>
> [9] Steikert, K. U., “The Weighted Nadaraya-Watson Estimator: Strong Consistency Results, Rates of Convergence, and a Local Bootstrap Procedure to Select the Bandwidth.” Diss. University of Zurich, 2014.
>
> [10] Hall, P., Wolff, R. C., and Yao, Q., Methods for estimating a conditional distribution function. *Journal of the American Statistical association*, 94(445), 154-163, 1999.

---

> ### Author Response · Authors · 2024-11-24
>
> **Question 2&3: baseline methods**
>
> We appreciate your recommendations to make our list of baseline methods more comprehensive. Below, we provide a very brief summary of the methods used in our comparison:
>
> - Nadaraya-Watson estimator (Plain NW) [11, 12]: Serves as a baseline where weights are derived directly from the original NW estimator without reweighting.
> - Sequential Predictive Conformal Inference (SPCI) [13]: Predicts quantiles of future residuals using quantile regression, with an adaptive mechanism to update estimates as new data becomes available.
> - Ensemble Prediction Interval (EnbPI) [14]: Constructs intervals by leveraging ensemble learners and building confidence intervals based on empirical quantiles.
> - Adaptive Conformal Inference (ACI) [15] and Aggregated ACI (AgACI) [16]: Dynamically adjusts the significance level for interval calibration based on historical coverage information, using empirical quantiles for prediction intervals. AgACI adapts online expert aggregation to avoid choosing the learning rate.
> - Fully Adaptive Conformal Inference (FACI) [17]: Extends ACI by tuning the learning rate over time.
> - Strongly Adaptive Online Conformal Prediction (SAOCP) [18]: Adapts strongly adaptive regret minimization and scale-free algorithms to the problem of online prediction by managing multiple experts, where each expert represents an individual online learning algorithm.
> - Scale-Free Online Gradient Descent (SF-OGD) [18, 19]: Strong regret minimization algorithm which can function as an independent algorithm for online conformal prediction.
> - Split Conformal Prediction (SCP) [20]: Foundational approach that leverages a calibration set.
>
> The methods can be broadly categorized into two main approaches for online conformal prediction:
> - Threshold-based updates: Methods such as ACI, AgACI, FACI, and SF-OGD focus on adaptively updating the threshold or significance level that determines the prediction sets at each time step.
> - Quantile-based prediction: SPCI and KOWCPI, on the other hand, cast the problem as predicting the conditional quantile of future residuals and adaptively refining these estimates. This difference in approach highlights KOWCPI’s distinct focus on quantile prediction for enhanced flexibility and accuracy.
>
> We sincerely appreciate the recommendations for additional methods including PID, BCI, and decay-OGD. In the revision, we have included references to the suggested methods to provide a more comprehensive discussion of related works. We also commit to conducting experiments to incorporate comparisons with these methods. Furthermore, we will include the implementation details and algorithms for every method in the Appendix of the final revision, ensuring clarity and transparency beyond the brief summaries in the main text.
>
> [11] Nadaraya, E. A., On Estimating Regression. *Theory of Probability & Its Applications*, 9(1):141–142, 1964.
>
> [12] Watson, G. S., Smooth regression analysis. *Sankhyā: The Indian Journal of Statistics, Series A*, 26:359–372, 1964.
>
> [13] Xu, C. and Xie, Y., Sequential Predictive Conformal Inference for Time Series. In *Proceedings of
> the 40th International Conference on Machine Learning*, 2023.
>
> [14] Xu, C. and Xie, Y., Conformal prediction for time series. *IEEE Transactions on Pattern Analysis
> and Machine Intelligence*, 45(10):11575–11587, 2023.
>
> [15] Gibbs, I. and Candès, E., Adaptive Conformal Inference Under Distribution Shift. *Advances in Neural Information Processing Systems*, 34:1660–1672, 2021.
>
> [16] Zaffran, M., Féron, O., Goude, Y., Julie Josse, and Aymeric Dieuleveut., Adaptive
> Conformal Predictions for Time Series. In *Proceedings of the 39th International Conference on
> Machine Learning*, 2022.
>
> [17] Gibbs, I. and Candès, E., Conformal Inference for Online Prediction with Arbitrary Distribution Shifts. *Journal of Machine Learning Research*, 25(162):1–36, 2024.
>
> [18] Bhatnagar, A., Wang, H., Xiong, C., and Bai, Y., Improved Online Conformal Prediction via Strongly Adaptive Online Learning. In *Proceedings of the 40th International Conference on Machine Learning*, 2023.
>
> [19] Orabona, F. and Pál, D., Scale-free online learning. *Theoretical Computer Science*, 716:50–69, 2018.
>
> [20] Vovk, V., Gammerman, A., and Shafer, G. *Algorithmic Learning in a Random World*. Springer, 2005.

---

> ### Author Response · Authors · 2024-11-24
>
> **Question 4: adaptive selection for $w$**
>
> As mentioned in the paper, we have considered adaptive window selection, where $w$ is not fixed but dynamically adjusted, using a two-sample Kolmogorov-Smirnov (KS) test. Specifically, at each time step $t$, we perform the two-sample KS test between two blocks of residuals: one comprising the most recent $w$ residuals of $(t - 1, \ldots, t - w)$ and the other comprising the $w$ residuals preceding that block in $(t - w - 1, \ldots, t - 2w)$. We then select the smallest $w$ value for which the p-value falls below 0.01. This approach is motivated to detect potential distributional changes in the residuals, akin to change point detection. Table 2 summarizes how the performance of KOWCPI on real datasets is affected by this approach.
>
> This approach allows for a data-driven and adaptive selection of $w$, removing the need for tuning $w$. Through experiments on the real data, we have confirmed that this method achieves comparable performance to $w$ values pre-selected by cross-validation. Although there is no significant enhancement in performance (but rather, it lost a bit), a key advantage is that $w$ is no longer a hyperparameter but determined directly from data.
>
> For further details, please refer to the newly added Appendix E in the rebuttal revision, where Table A.4 compares results for each dataset with and without adaptive window selection, and Figure A.3 visualizes the dynamic adjustment of $w$ at each time step. We acknowledge that this approach could be further refined, such as employing a formal change-point detection algorithm instead of the simple KS test. This refinement could potentially not only make selection of $w$ data-driven but also lead to improved performance compared to fixed $w$. We will polish this argument and include these considerations in the revision for greater clarity.
>
> Table 2: Comparison of KOWCPI on real datasets using pre-fixed window lengths selected by cross-validation versus adaptive window selection based on the two-sample KS test. Target coverage is 90%, and the standard deviations are derived across five independent trials.
> | Method         | Electric Coverage | Electric Width | Wind Coverage | Wind Width | Solar Coverage | Solar Width |
> |----------------|--------------------------|-----------------------|----------------------|-------------------|-----------------------|--------------------|
> | Fixed $w$      | 0.90 (2.3e-3)           | 0.23 (1.5e-3)        | 0.91 (2.8e-3)       | 2.41 (3.2e-2)    | 0.90 (1.2e-3)        | 48.8 (9.4e-1)     |
> | Adaptive $w$   | 0.92 (3.0e-3)           | 0.22 (1.3e-3)        | 0.90 (4.4e-3)       | 2.44 (2.7e-2)    | 0.90 (1.3e-3)        | 50.6 (1.1e0)      |

---

> ### Comment · Reviewer_75Ps · 2024-11-24
> **Response to the authors**
>
> Thank the authors for their effort in conducting additional experiments and adding related works. Most of my concerns have been addressed and I have raised my score. I notice that the revision exceeds the upper paper limit of 10 pages slightly and kindly remind the authors to revise it to less than 10 pages as soon as possible.

---

> > ### Author Response · Authors · 2024-11-24
> >
> > We sincerely appreciate your recognition and we are truly delighted to hear that most of your concerns have been addressed. Also, thank you for pointing out our mistake, and we have revised it to fit within the 10-page limit. Thank you again for your kind reminder and support!

---

### Official Review · Reviewer_LFjW · 2024-11-04

**Soundness:** 3
**Presentation:** 3
**Contribution:** 3
**Rating:** 6
**Confidence:** 3

**Summary:**

This article studies how to use the conformal prediction framework to establish prediction intervals for time series data.
The key point of the method is: to estimate the conditional distribution of future residuals using the Reweighted Nadaraya-Watson (RNW) method. Then establish the prediction interval for the future response with mean function and estimated distribution of residual. The experiments indicate that the method produces shorter and more efficient prediction intervals while meeting coverage requirements.

**Strengths:**

- This article applies the conformal prediction and the RNW method to establish a prediction interval. Therefore, the proposed method is independent of specific time series models and has a broader range of applications.

- Instead of establishing prediction intervals using the quantiles $Q_{\alpha/2}$ and $Q_{1-\alpha/2}$. The method seeks a $\beta$ such that the distance between the quantiles $Q_{\alpha/2+\beta}$ and $Q_{1-\alpha/2+\beta}$ is minimized to form the prediction intervals. This approach is more effective for certain residual distributions.

- The authors consider time-series structures. In conditional coverage properties, the authors loosen the stationary condition only on strongly mixing residuals.

- The experiments indicate that the method produces shorter and more efficient prediction intervals while meeting coverage requirements.

**Weaknesses:**

- The authors make a natural integration of the conformal prediction with the RNW method. However, the innovation of this approach is not particularly strong. The theoretical results primarily focus on the asymptotic accuracy of distribution estimation.

- When the dependence in the time series is strong, the dimension of $\tilde{X}$ may be large. In this case, when the data is limited, kernel estimates may not yield reliable results. Also, large $w$ leads to a low convergence rate of coverage. So it is better to include experimental results showing how coverage and interval width as $w$ increases for different sample sizes and datasets with varying degrees of temporal dependence.

-  The method establishes prediction intervals for stationary time series. Can the method be extended to handle local stationarity? Have authors considered techniques like differencing or detrending to handle non-stationarity? Additionally, please provide a more explicit discussion of the limitations of the proposed method for non-stationary data.

**Questions:**

-  $T$ represents the number of past residuals used to give the estimation of residual at the next time point. Can the authors give particular analyses related to $T$, such as a sensitivity analysis showing how prediction interval performance changes with different values of $T$, or a discussion of how $T$ was chosen for each dataset?

- The method involves two important parameters: the bandwidth $h$ and the window length $w$. While the paper discusses the settings for these parameters, it is unclear whether the experimental results are sensitive to them. Additional experimental results may be needed to clarify this.

- When the dimension of $\tilde{X}$ is large, the results from kernel estimation may not be robust. In this case, is it possible to train a function to compress $\tilde{X}$, aiming to extract effective information and reduce dimensionality for $\tilde{X}$?

- In classic time series models like the ARMA-GARCH model, can the choice of $w$ or $h$ be discussed more precisely in theoretical parts as a corollary?

- Can the authors provide some experimental results on non-stationary time series?

---

> ### Author Response · Authors · 2024-11-23
>
> Dear Reviewer LFjW,
>
> We greatly appreciate your comments and recommendations. We have structured our responses to ensure that all your concerns are addressed clearly and comprehensively.
>
> **Weakness 1: innovations of KOWCPI & asymptotic results**
>
> Our primary innovation lies in adapting the RNW estimator, a method specifically designed for CDF/quantile estimation in time-series data, to construct adaptive weighting for weighted conformal prediction with streaming data. The RNW estimator is particularly advantageous because it reduces bias and directly provides a valid CDF, making it an ideal fit for a quantile-based approach to conformal prediction. This alignment also allows us to develop natural theoretical results based on the RNW estimator’s consistency and convergence rate.
>
> Our experiments confirm the effectiveness of this approach, demonstrating that KOWCPI achieves superior performance in practice compared to existing methods. By leveraging the reweighting mechanism of the RNW estimator, KOWCPI adapts well to data dynamics, providing narrower confidence intervals while maintaining valid coverage. Please refer to Lines 474-485 in the Experiments section of the revision, where we have added a discussion on why this reweighting mechanism plays a crucial role in achieving success in online conformal prediction.
>
> **Discussion about asymptotic results**
>
> We acknowledge that the conditional coverage results presented are limited to being asymptotic. However, without exchangeability of data, in the non-asymptotic case, even for marginal coverage, the most general approach is to provide a bound on the coverage gap [5, 6]. Therefore, our asymptotic results remain helpful and somewhat inevitable. For conditional coverage, there was inherent difficulty in providing an explicit nonasymptotic bound for the gap, rising from the fact that the weights of the RNW estimator are not martingale, with its uniform decay not known in literature, to the best of our knowledge.
>
> Despite the inherent difficulties, we attempted to provide bounds on the marginal coverage gap using a permutation-based argument. While expressing the bound in terms of the distance between weights might seem somewhat abstract, this approach naturally arises when dealing with non-exchangeable data. Similar approaches have been utilized in other seminal works, including [5, 7].
>
> **Weakness 2 and Question 2: sensitivity to $h$ and $w$**
>
> Indeed, the wrongful selection of the bandwidth $h$ can result in either poor coverage or overly conservative confidence intervals. Specifically, smaller $h$ values often lead to insufficient coverage, while larger $h$ values tend to cause over-coverage and excessively wide intervals. To illustrate this sensitivity, we conducted an additional experiment using the Wind dataset. Instead of using the nonparametric AIC, which was the method employed in the paper for bandwidth tuning, we randomly selected $h$ values within the range $(1, 10)$ for 100 trials. The results show that with random bandwidth selection:
> - Mean coverage: 0.93 (std 3.1e-2)
> - Mean width: 4.03 (std 1.0e0)
>
> In comparison, the original method using nonparametric AIC produced:
> - Mean coverage: 0.91 (std 2.8e-3)
> - Mean width: 2.41 (std 3.2e-2)
>
> This simple experiment demonstrates that random selection of $h$ leads to significantly wider intervals and a much larger variability in both coverage and width. This highlights the critical importance of bandwidth tuning. The nonparametric AIC approach used in our paper ensures a principled and consistent selection of $h$, resulting in narrower intervals and more stable results while maintaining valid coverage.
>
> For $w$, which we proposed to have less impact on the results compared to $h$, we conducted a similar experiment. In this case, $w$ was randomly selected within a range of 1 to 30. The results are as follows:
> - Mean coverage: 0.91 (std 1.5e-3)
> - Mean width: 2.45 (std 4.5e-2)
>
> The results remain close to those obtained with our default $w$ selection via cross-validation, still achieving the narrowest interval among baseline methods with valid coverage. This implies that while $w$ can influence performance to some extent, it is less critical than bandwidth tuning. (Please note that bandwidth $h$ is tuned via nonparametric AIC for each selected $w$.)
>
>
> [5] Barber, R. F., Candes, E. J., Ramdas, A., and Tibshirani, R. J., Conformal prediction beyond exchangeability. *The Annals of Statistics*, 51(2): 816-845, 2023.
>
> [6] Lei, J., and Wasserman, L., Distribution-free prediction bands for non-parametric regression. *Journal of the Royal Statistical Society Series B: Statistical Methodology* 76.1: 71-96, 2014.
>
> [7] Tibshirani, R. J., Barber, R. F., Candes, E., & Ramdas, A., Conformal prediction under covariate shift. *Advances in Neural Information Processing Systems*, 32, 2019.

---

> ### Author Response · Authors · 2024-11-23
>
> **Weakness 3 and Question 5: non-stationary data**
>
> - We appreciate your thoughts on the applicability of KOWCPI to non-stationary data. To clarify, as stated in the paper and reiterated in Point 1 of the common response, our approach does not require the **original time series data** to be stationary. Instead, the theoretical guarantees on conditional coverage rely on the stationarity and strong-mixing assumptions of the **residuals**.
> - However, to illustrate an example, we note that the Wind dataset used in our experiments indeed is suggested to be non-stationary based on the ADF test, but KOWCPI successfully demonstrated robust performance.
> - To further illustrate its applicability, we conducted additional simulations using non-stationary synthetic data, and included the specifics and the results in newly added Appendix D. Across all scenarios, according to Table A.2 and A.3, KOWCPI achieved the narrowest confidence intervals in general with valid coverage compared to baseline methods, demonstrating its robustness in non-stationary settings.
>
> **Question 1: analysis on $T$**
>
> For our experiments, we utilized the full training data to train the quantile estimator. If the stationarity assumptions of the residuals hold, a larger $T$, representing the training data size used for the quantile estimator (precisely speaking, $n = T-w$), will generally lead to better performance due to more accurate estimation. However, in scenarios involving nonstationarity or distribution shifts, it is possible that using excessively old residuals may degrade the estimation quality. To explore this further, we conducted additional experiments with different $T$s on the Solar dataset. The results in Table 1 show that larger $T$ values generally lead to narrower intervals and better performance as expected.
>
> Table 1: Results for Solar dataset with different choices of $T$. Target coverage is 90%. Standard deviations are derived across five independent trials.
> | T     | Coverage        | Width         |
> |-------|------------------|---------------|
> | 1600  | 0.90 (1.2e-3)   | 48.8 (9.4e-1) |
> | 800   | 0.90 (2.7e-3)   | 49.0 (1.1e0)  |
> | 400   | 0.90 (1.8e-3)   | 49.3 (8.8e0)  |
> | 200   | 0.89 (6.8e-3)   | 53.4 (1.8e0)  |
> | 100   | 0.89 (1.3e-2)   | 72.4 (2.3e0)  |
>
> **Question 3: dimensionality reduction of $\tilde X$**
>
> Thank you for this suggestion. We agree that dimension-reduction techniques, such as PCA or projected kernels, can be applied to $\tilde{X}$ to address potential issues with high dimensionality. We will conduct experiments for these approaches and include the results in the final revision if they show promise. That said, as demonstrated in our experiments across various datasets, it is worth noting that KOWCPI has consistently performed effectively even without applying dimensionality reduction techniques.
>
> **Question 4: theory with ARMA-GARCH model**
>
> In theory, for a given $w$, the optimal bandwidth $h$ that minimizes the asymptotic mean integrated squared error is well-discussed in the literature: $h \propto n^{-1/(w+4)}$, where the proportionality constant depends on the kernel function and the distribution of the data [8]. However, how exactly this proportionality constant is influenced by the specific distributions arising in ARMA-GARCH models is not fully understood. This is an intriguing topic, and we will consider including a discussion if we identify meaningful theoretical insights.
>
> While we cannot directly connect the choices of $w$ and $h$ to ARMA-GARCH parameters in theory, we conducted experiments using an ARMA-GARCH mixture model. KOWCPI achieved favorable results by selecting $w$ and $h$ via cross-validation and nonparametric AIC as describe in the paper. The details of these experiments have been added to the revision in Appendix D.1.
>
> [8] Wand, M. P., and Jones, M.C., *Kernel smoothing*. CRC press, 1994.

---

> > ### Comment · Reviewer_LFjW · 2024-11-27
> >
> > Thank you for the detailed response. I will keep my score and tend to accept.

---

> > > ### Author Response · Authors · 2024-11-27
> > >
> > > We thank the reviewer for the confirmation and positive recommendation!

---

### Official Review · Reviewer_E4WM · 2024-11-04

**Soundness:** 3
**Presentation:** 3
**Contribution:** 3
**Rating:** 6
**Confidence:** 3

**Summary:**

The authors introduce KOWCPI to address the difficulties that arise from employing conformal prediction in the time series setting. In order for standard conformal prediction procedures to have valid statistical coverage guarantees the data needs to be exchangeable. However, this requirement is violated in time series data. KOWCPI uses the Reweighted Nadaraya-Watson estimator for quantile regression in the presence of dependent data to learn the distribution of residuals. Using this information, intervals can be constructed that have been theoretically shown to provide conditional coverage and empirically have been found to be narrower than other baselines while maintaining desired coverage.

**Strengths:**

- Authors provide a conditional coverage guarantee (much more meaningful than marginal coverage guarantee)
- Proofs are included and assumptions are clearly stated
- Strong coverage and efficiency results on intervals

**Weaknesses:**

- Grammatical errors/typos scattered throughout
- Notation is rather dense. More plain language accompanying the mathematical formalism would help the reader
- Presentation of Algorithm 1 seems to be missing some pieces. In the required line should it be $(X_t, Y_t), t=1, 2, ..., T$? Variables $n, w$ are referenced but not explained earlier in the box.
- Table 1 is a little bit confusing at first glance. Typically bold indicates "best" but in this case it only shows what the authors' method is
- See questions

**Questions:**

- Is quantile crossing ever an issue in $\hat Q_\beta$?
- The form of the intervals is $\hat C^\alpha_{t-1} = [\hat f(X_T) + \hat Q_{\beta^*}(\tilde X_{n+1}) , \hat f(x) + \hat Q_{1 - \alpha + \beta^*}(\tilde X_{n+1})]$. Are there cases where both quantile estimates are positive/negative and possibly leading to invalid intervals?
- Are there ever discontinuities over time? (ie the quantile estimators dramatically jump around)
- Are there any diagnostics that can be used to assess the plausibility of the assumptions?
- How robust is this method to violation of assumptions?

---

> ### Author Response · Authors · 2024-11-23
>
> Dear Reviewer E4WM,
>
> We sincerely appreciate your thoughtful review. In the following, we will provide a comprehensive response to your feedback.
>
> **Weaknesses 1 and 2: typos and dense notation**
>
> We appreciate your attention to detail. We recognize that some parts were grammatically awkward, and some were notation-heavy. We have carefully reviewed the paper and addressed these issues in the revision. We have double-checked whether there is any undefined notations, and have added explanations for notations and equations in:
> - Eq. (7): The final weights of the RNW estimator,
> - Proposition 4.6: Defined $D_{\tilde x}^2$ (the Hessian), which was originally missing.
> - Proof of Lemma B.2: Briefly explained definitions of $J_1, J_2, J_3$.
>
> **Weakness 3: Algorithm 1 presentation**
>
> Thank you for pointing out the ambiguity in Algorithm 1. The original intent of not terminating the index at $T$ was to reflect the online nature of the algorithm, allowing for continuous updates as new data becomes available. However, we recognize that it is clearer to explicitly state that the training data is provided up to $T$, and the algorithm constructs prediction intervals for $t = T+1, T+2, \ldots$. In response, we have revised Algorithm 1 to reflect this structure and to explicitly define all variables, such as $n$ (number of the residual blocks used to train) and $w$ (window length), and also polished the plain language explanations for clarity.
>
> **Weakness 4: highlighting in Table 1**
>
> We apologize for the confusion that the table mistakenly used bold text to highlight only our method (KOWCPI) rather than the best-performing methods. In the revision, we have removed highlights from the table to avoid any potential misinterpretation.
>
> **Question 1: quantile crossing**
>
> No, quantile crossing does not happen with our approach. The RNW estimator for the conditional CDF, $\hat{F}$, is a monotone increasing step function by construction. Therefore, the quantile estimator $\hat{Q}$, defined as the generalized inverse of $\hat{F}$, inherently satisfies $\beta_1 > \beta_2 \implies \hat{Q}\_{\beta_1} \geq \hat{Q}\_{\beta_2}$.
>
> **Question 2: upper/lower quantile signs and validity**
>
> Yes, it is possible for both quantile estimates to have the same sign for a given $\alpha$ when the residuals are skewed or not centered around zero. However, this does not render the interval invalid. In fact, it reflects an advantage of KOWCPI’s confidence interval, which can correct biases in the point prediction.
>
> When the residuals are consistently large in one direction (positive or negative), it suggests that the underlying point prediction algorithm is biased. In such cases, our confidence interval compensates for this by adjusting the interval to more accurately reflect the true distribution, rather than centering around a potentially biased point estimate. This effectively “fixes” the initial prediction bias by providing an interval that includes the correct quantile range.
>
> Indeed, in our experiments, we frequently observed this behavior, where the confidence intervals adapted with the same sign for both the lower and upper bounds to correct biased point predictions, ultimately resulting in accurate interval coverage. For example, in this [Figure](https://anonymous.4open.science/r/KOWCPI_figures-97BD/quantile_signs.png) (anonymous link), which zooms on a portion of the prediction for the Electric dataset, after index 50, where the bias in the point prediction is large, the blue region of confidence interval no longer contains the yellow line representing $\hat{Y}$. In this case, both the upper and lower quantiles have the same sign due to the large bias in the point prediction, leading to successful coverage.

---

> ### Author Response · Authors · 2024-11-23
>
> **Question 3: discontinuity over time**
>
> Thank you for raising this point. If the question refers to abrupt changes in the quantile estimates over time, the answer is yes, and we believe this is one of the primary reasons behind KOWCPI’s success. Unlike methods using empirical quantiles or relying on fixed weights, KOWCPI’s adaptive mechanism, particularly its kernel-based reweighting, enables it to respond quickly to shifts in data.
>
> This adaptability allows KOWCPI to effectively track sudden changes in the data trajectory, ensuring that the prediction intervals remain relevant and accurate, even in highly volatile scenarios. For example, in Appendix D.1 of the rebuttal revision, we present simulations where KOWCPI quickly adjusts to volatility bursts, producing intervals that dynamically follow the true trajectory while avoiding over-coverage.
>
> While this adaptability is a strength, we acknowledge that it may introduce challenges in certain edge cases, such as when the data stabilizes immediately after a burst. However, our experiments suggest that KOWCPI generally balances this trade-off effectively, maintaining both coverage validity and narrow interval widths across a variety of datasets. Please also refer to Lines 474-485 in the Experiments section of the revision, where we added a detailed discussion in this regard.
>
> **Question 4: diagnostics for assumptions on data**
>
> We have two assumptions on data, Assumptions 4.2 and 4.3, while 4.4 and 4.5 pertaining to the choice of the kernel function and bandwidth.
>
> To evaluate Assumption 4.2, which concerns the $\alpha$-mixing properties of the residuals and the decay of the mixing coefficient, one approach is to use the goodness-of-fit test described in [4]. This formal statistical test verifies whether the $\alpha$-mixing coefficient decays faster than a specified rate. Additionally, ACF or PACF plots of the residuals can provide empirical evidence about the dependency structure and help diagnose whether strong mixing assumptions are likely to hold.
>
> For Assumption 4.3, which pertains to the smoothness of conditional CDF and densities, we believe that exploratory data analysis, such as histograms of the residuals or summary statistics, can provide empirical insights into whether this assumption holds. If we identify a formal testing method for verifying Assumption 4.3, we will include this discussion in the final revision.
>
> **Question 5: robustness to assumption violations**
>
> While our theoretical guarantees rely on the assumption of stationary residuals, our method has demonstrated strong empirical performance even when this assumption is partially violated. For instance, the residuals in the Wind dataset are suggested to be nonstationary according to the ADF test (please note that this differs from what we mentioned in Point 1 of the common response; both the Wind data and its residuals exhibit nonstationarity. See [Figure](https://anonymous.4open.science/r/KOWCPI_figures-97BD/residual_plot.png) for the residual plot). Still, KOWCPI consistently achieves the narrowest interval width while maintaining valid coverage. Additionally, we conducted new experiments with synthetic data exhibiting nonstationary behavior in the residuals. Still in these scenarios, KOWCPI generally outperformed baseline methods. For further details, please refer to Appendix D and Point 3 of the common response.
>
> [4] Khaleghi, A. and Lugosi, G., Inferring the Mixing Properties of a Stationary Ergodic Process From a Single Sample-Path. In *IEEE Transactions on Information Theory*, vol. 69, no. 6, pp. 4014-4026, 2023.

---

> > ### Comment · Reviewer_E4WM · 2024-11-25
> >
> > Thank you for the reply. Table 1 still looks a little awkward in terms of formatting (minor). I will update my score.

---

> > > ### Author Response · Authors · 2024-11-25
> > >
> > > We truly appreciate your thoughtful feedback and your recognition of our revision. Regarding Table 1, we have refined its formatting in the latest revision to address your concern. Your attention to detail is greatly valued, and thank you again for your support!

---

### Author Response · Authors · 2024-11-23
**Common response to all reviewers**

Dear Reviewers,

We sincerely appreciate the time and effort you dedicated to providing constructive feedback, as well as your recognition of our paper’s strengths. In the common response, we aim to address feedback points common among reviewers and highlight key aspects of our approach that we believe are important to emphasize for all reviewers. Full responses to each specific review are provided in the individual comments.

**Point 1. Strong mixing assumptions on residuals**

We first emphasize that we do **NOT** assume stationarity or strong-mixing of the original time series data $(X_t, Y_t)$. Instead, our theoretical results on conditional coverage rely on the stationarity and strong-mixing assumptions of the non-conformity scores (residuals). As clarified in Lines 320-323, this requirement is far less restrictive than assuming stationarity or strong mixing of the original series. It can be satisfied even when the data is clearly non-stationary or lacks mixing, e.g., a vector AR model with a time-dependent drift.

We also note that the Wind dataset used in our experiments is proposed to be non-stationary by the ADF test. As elaborated below on Point 3, we have also conducted additional experiments to further evaluate KOWCPI’s performance under non-stationary conditions.

**Point 2. Hyperparameter tuning**

Our theoretical results hold for any choice of window length (dimension of $\tilde X$) $w$, with a corresponding choice of bandwidth $h$ based on Assumption 4.5. Theoretically, optimal bandwidth selection is studied well in the literature of nonparametric estimation, and we utilized such theoretical choices of optimal bandwidth for constructing our theoretical results.

However, in practice, $w$ and $h$ become hyperparameters that require tuning. Bandwidth selection is critical for performance, and we have employed nonparametric AIC, a commonly used method in the literature for tuning the bandwidth of linear smoothers ([1]). In response to shared concerns about tuning $w$, we have suggested the implementation of an adaptive selection process for $w$, using a two-sample Kolmogorov-Smirnov test, motivated by the intention to exploit the distribution of residuals. Rather than relying on tuning via cross-validation to fix $w$ in prior, this can ensure $w$ to be chosen in a data-driven manner, with a minimal loss in performance in practice. Please refer to the newly added Appendix E in the revision for a more detailed discussion.

**Point 3. Additional experiments**

Furthermore, to address your shared recommendations on experiments, we conducted additional experiments, including:
- (Real data) Apple’s daily closing stock prices,
- (Synthetic data 1) A heteroskedastic mixture model of AR(1) + GARCH(1,1) given as
$$
	Y_t = 0.8Y_{t-1} + \sigma_t \epsilon_t + \xi_t, \quad \sigma_t^2 = 0.1 + 0.3Y_{t-1}^2 + 0.6 \sigma_{t-1}^2, \quad \epsilon_t \stackrel{iid} \sim N(0,1), \quad \xi_t \stackrel{iid} \sim N(0, 0.1^2),
$$
- (Synthetic data 2) A nonstationary time series with a seasonal trend given as
$$
	Y_t = \log(t’) \sin \left(\frac{2 \pi t’}{12} \right) (|\beta^T X_t| + |\beta^T X_t|^2 + |\beta^T X_t|^3)^{1/4} + \epsilon_t, \quad \epsilon_t \sim AR(1).
$$
We consistently observe the coverage validity of KOWCPI on these diverse datasets, and KOWCPI yields the shortest intervals on average. For further details, please refer to Tables A.1–A.3 and Appendices C and D, newly added to the rebuttal revision.

Additional experiments in individual responses include:
- Selection of $T$ and its impact on performance.
- Adaptive selection of $w$ using the two-sample Kolmogorov-Smirnov test.
- Real data experiments with different target coverage levels.

**Changes in rebuttal revision**

In the rebuttal revision, we have made the following changes to address reviewer feedback:
- (Lines 474-485 on experiment results) Added qualitative reasoning for superior performance of KOWCPI pertaining to the reweighting.
- Added detailed descriptions of the additional experiments in Appendices C-D.
- Added discussion and experiments about adaptive window selection in Appendix E.
- Removed misleading highlights in Table 1 and Figure 2.
- Added standard deviations for coverage and width terms in the experimental results, including Table 1.
- Revised Algorithm 1 for improved clarity.
- Included references to the suggested works ([2], [3]).
- Added plain text explanation for some notations.
- Corrected typos and grammar.

Changes in the revision are highlighted in blue.

[1] Cai, Z., Regression quantiles for time series. *Econometric Theory*, 18(1):169–192, 2002.

[2] Yang, Z., Candès, E., and Lei, L., Bellman conformal inference: Calibrating prediction intervals for time series. *arXiv preprint arXiv*:2402.05203, 2024.

[3] Angelopoulos, A. N., Barber, R. F., and Bates, S., Online conformal prediction with decaying step sizes. In *Proceedings of the 41st International Conference on Machine Learning*, 2024.

---

### Meta-Review · Area_Chair_ft1a · 2024-12-21

**Metareview:**

This work introduces a novel conformal prediction method for time-series, called Kernel-based Optimally Weighted Conformal Prediction Intervals (KOWCPI). KOWCPI adapts the Reweighted Nadaraya-Watson (RNW) estimator for quantile regression on dependent data and learns optimal data-adaptive weights. The paper addresses the challenge of establishing a conditional coverage guarantee for non-exchangeable data under strong mixing conditions on non-conformity scores. Experimental results show that KOWCPI outperforms state-of-the-art methods on real time-series, achieving narrower confidence intervals without compromising coverage. After the author response and author-reviewer discussions, this paper has received unanimous support from the reviewers. Therefore, I recommend acceptance.

**Additional Comments On Reviewer Discussion:**

The original reviews raised several concerns, including typos, unclear writing, insufficient experiments, and missing related work. The authors did an excellent job addressing these issues in their rebuttal, leading to increased scores from three reviewers (E4WM, 75Ps, WMCc).

---

### Decision · Program_Chairs · 2025-01-22

Accept (Poster)